# DIFFERENTIABLE CLUSTER GRAPH NEURAL NETWORK

## ABSTRACT

Graph Neural Networks often struggle with long-range information propagation and local heterophilous neighborhood aggregation. Inspired by the observation that cluster patterns manifest at global and local levels, we propose to tackle both challenges with a unified framework that incorporates a clustering inductive bias into the message passing mechanism, using additional cluster-nodes. Central to our approach is the formulation of an optimal transport based clustering objective. However, optimizing this objective in a differentiable way is non-trivial. To navigate this, we adopt an iterative process, alternating between solving for the cluster assignments and updating the node/cluster-node embeddings. Notably, our derived optimization steps are themselves simple yet elegant message passing steps operating seamlessly on a bipartite graph of nodes and cluster-nodes. Our clustering-based approach can effectively capture both local and global information, demonstrated by extensive experiments on heterophilous and homophilous datasets.

## 1 INTRODUCTION

Graph Neural Networks (GNNs) have emerged as prominent models for learning node representations on graph-structured data. Their architectures predominantly adhere to the message passing paradigm, where node embeddings are iteratively refined using features from its adjacent neighbors (Kipf and Welling, 2016; Defferrard et al., 2016; Gilmer et al., 2017). While this message passing paradigm has proven effective in numerous applications (Ying et al., 2018; Zhou et al., 2020), two prominent challenges have been observed. First, long-range information propagation over a sparse graph can be challenging (Li et al., 2018; Zhou et al., 2021; Rusch et al., 2023). Expanding the network's reach by increasing the number of layers is often suboptimal as it could encounter issues such as over-squashing (Alon and Yahav, 2020; Topping et al., 2021), where valuable long-range information gets diluted as it passes through the graph's bottlenecks, diminishing its impact on the target nodes. Second, some graphs exhibit heterophily, where connected nodes are likely to be dissimilar. In such cases, aggregating information from the dissimilar neighbors might introduce noise and hinder the graph representation learning performance (Zhu et al., 2020b; 2021).

In this paper, we focus on the task of supervised node classification using GNN and explore clustering as an inductive bias to address both challenges. Our approach is motivated by the observation that cluster patterns can be utilized at both global and local levels in graph data. Globally, cluster patterns appear when nodes that are far apart in the graph exhibit similar features (see Fig. 1). These patterns can be leveraged to enable efficient long-range information transfer, by clustering nodes by their latent space representations, rather than structural proximity. Locally, particularly for heterophilic neighbourhoods, it would be desirable to disconnect edges across dissimilar nodes while maintaining connections across similar nodes. Clustering nodes within local neighbourhoods provides a mechanism for this.

To embed the clustering inductive bias explicitly into the network architecture, we propose Differentiable Cluster Graph Neural Network (DC-GNN), an efficient end-to-end learning framework designed to address both over-squashing and heterophily. We first formulate the problem of inferring cluster-aware node representations as an optimization task with a novel clustering-based objective function. This objective function is composed of a weighted sum of global and local clustering terms, which promote long-range information propagation and effective aggregation in heterophilic neighborhoods, respectively. A key challenge in integrating clustering into an end-to-end learning framework

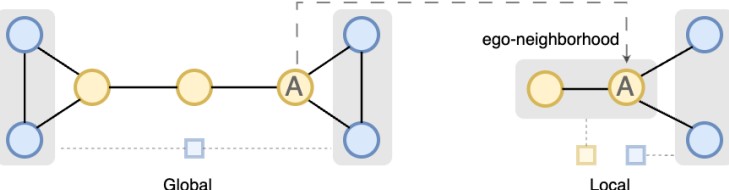

Figure 1: An illustration of global and local cluster patterns, where nodes of the same color share similar features and square boxes indicate conceptual cluster centroids. On the left is an instance where distant blue nodes are similar and get connected via global cluster centroid. On the right is the heterophilous ego-neighborhood of node A where similar nodes connected by their respective cluster centroids.

lies in the non-differentiability of classical clustering algorithms. To achieve differentiability, we define the objective function that seeks to move node representations to their cluster-centroids through the lens of an Optimal Transport (OT) problem (Villani et al., 2009). Unfortunately, optimizing this clustering-based objective function as a conventional training loss is inherently difficult due to the latent nature of cluster assignments. To overcome this, we propose an alternating optimization approach based on block coordinate descent which iterates between (1) solving for a soft cluster assignment matrix that probabilistically assigns a node to a cluster, and (2) updating the node/cluster embeddings given the cluster assignment matrix.

Intriguingly, *this iterative alternating optimization algorithm for minimizing the clustering-based objective function can be interpreted as an iterative message passing procedure.* It operates on a bipartite graph consisting of original nodes and cluster nodes representing centroids. Unlike previous approaches that treat clustering as a separate component, our method directly embeds the clustering process into the message passing network architecture. This ensures the clustering-based objective function is optimized as part of the message passing network itself, enforcing the clustering inductive bias during both training and inference. The resulting cluster-aware node embeddings are then fed into a task-specific loss for supervised node classification, allowing DC-GNN to be trained end-to-end.

DC-GNN is efficient and has a linear complexity with respect to the graph size. Additionally, our framework can be viewed as a form of graph rewiring, where the introduction of cluster nodes creates new pathways between original nodes. This rewiring reduces the overall graph's effective resistance, helping to mitigate oversquashing (Black et al., 2023), as shown in our experiments. To assess the effectiveness of DC-GNN, we conduct extensive evaluations on 14 datasets, spanning both heterophilous and homophilous graphs. Our results demonstrate that DC-GNN consistently achieves superior or competitive performance compared to state-of-the-arts.

## 2 RELATED WORK

Prominent GNN models typically follow a message passing paradigm that iteratively aggregates information in a node's neighborhood (Kipf and Welling, 2016; Bruna et al., 2013; Defferrard et al., 2016; Gilmer et al., 2017; Veličković et al., 2017; Xu et al., 2018). This local message passing, however, requires the stacking of multiple layers to pursue long-range information and can encounter issues such as over-smoothing (Li et al., 2018; Cai and Wang, 2020; Rusch et al., 2023) and over-squashing (Alon and Yahav, 2020; Topping et al., 2021; Banerjee et al., 2022; Karhadkar et al., 2022). To tackle oversquashing, most existing works design graph rewiring techniques that change graph topology (Topping et al., 2021; Nguyen et al., 2023; Arnaiz-Rodríguez et al., 2022; Karhadkar et al., 2022). Other works like Chen et al. (2024) leverages random walk sequences and Kosmala et al. (2023) leverages Fourier basis representation. Similar to some existing methods (Black et al., 2023), our approach of adding cluster-nodes also changes graph topology and is shown to reduce effective resistance, thereby helpful in mitigating oversquashing (Black et al., 2023).

Additionally, some graphs contain heterophilous neighborhoods, in which traditional aggregation promoting similarity among neighbors is suboptimal. (Zhu et al., 2020b; 2021). There are three types of approaches to address this, including design of high-pass filters in message passing (Chien et al., 2020; Fu et al., 2022; Dong et al., 2021), exploring global neighborhoods (Xu et al., 2022; Jin et al., 2021; Li et al., 2022; Abu-El-Haija et al., 2019), and use of auxiliary graph structures (Pei et al., 2020; Zhu et al., 2020a; Lim et al., 2021; Yan et al., 2021). Many of these approaches are often

computationally expensive and may struggle on homophilous graphs. Our method seeks to explore global neighborhoods by introducing cluster-nodes as auxiliary structures and conduct clustered aggregation in local neighborhoods, with linear complexity.

Our work focuses on using a clustering inductive bias to enhance the supervised node classification task. This differs from tasks like graph clustering (Tsitsulin et al., 2023; Tian et al., 2014) and graph pooling (Bianchi et al., 2020; Duval and Malliaros, 2022), which are constrained by graph typology and aim to partition a graph into substructures. Instead, our approach primarily utilizes feature information for global clustering. Unlike previous methods, we also explore clustering within local neighborhoods, which has not been explored before.

Clustering within a differentiable pipeline has been explored particularly in unsupervised and self-supervised settings (Feng et al., 2022; Saha et al., 2023; Caron et al., 2018; Stewart et al., 2024). However, most of these approaches do not extend to graph-structured data. Few methods, such as DCAT (Zhou et al., 2024), have applied differentiable clustering in graph-based models, typically employing clustering as an auxiliary loss function. In contrast, our approach directly integrates clustering into the message-passing mechanism. By deriving the message-passing steps as optimization steps toward a clustering objective, we effectively embed a clustering algorithm into the model's architecture. This design enables clustering to be performed both during training and inference.

Our approach, which integrates clustering into message passing, is realized through an Optimal Transport-based clustering objective. OT has been recently applied to graph learning for tasks like graph classification (Titouan et al., 2019; Bécigneul et al., 2020; Vincent-Cuaz et al., 2022; Ma et al., 2024), regularizing node representations (Yang et al., 2020) and finetuning (Li et al., 2020). However, most of these methods leverage OT as a separate component and do not integrate it within message passing. Distinct from these approaches, we implicitly optimize an OT-based clustering objective function via message passing.

## 3 METHODOLOGY

In this section, we introduce our unified clustering-based GNN message passing method to address both long-range interactions and heterophilous neighborhood aggregation. This involves transforming the input graph into a bipartite graph by introducing cluster-nodes, defining an optimal transport (OT) based clustering objective function, and optimizing it through our derived message passing steps within a differentiable coordinate descent framework. Notations are introduced where needed throughout the paper. A complete list of these notations is available in Appendix A.

### 3.1 DC-GNN FORMULATION

We begin by constructing a bipartite graph, denoted as $\mathcal{G} = (\mathcal{V}, \mathcal{C}, \mathcal{E})$. This bipartite graph is derived from the original graph $G = (V, E)$ and comprises two distinct sets of nodes. The first set, $\mathcal{V}$, is a direct copy of the nodes $V$ from the original graph. The second set, $\mathcal{C}$, consists of cluster-nodes divided into two categories: global clusters ($\Omega$) and local clusters ($\Gamma$). $\mathcal{E}$ and $E$ represent the set of edges in the bipartitie graph and original graph respectively.

In this bipartite graph, each global cluster-node from $\Omega$ connects to all nodes in $\mathcal{V}$, thereby facilitating long-range interactions across distant nodes. Meanwhile, each local cluster-node from $\Gamma$ is associated with a specific node $i$ in $\mathcal{V}$, and connected to its ego-neighborhood. The ego-neighborhood of node $i$, denoted as $\mathcal{N}_i^+$, includes $i$ itself and its one-hop neighbors, formally defined as $\mathcal{N}_i^+ = \mathcal{N}_i \cup \{i\}$, where $\mathcal{N}_i = \{j : (i,j) \in E\}$. Here, $\Gamma_i$ represents the set of local clusters associated with node $i$ in $\mathcal{V}$. The total number of nodes in $\mathcal{C}$ equals the nodes in $\Omega$ plus those in all local clusters, i.e., $|\mathcal{C}| = |\Omega| + \sum_{i \in \mathcal{V}} |\Gamma_i|$. An illustration of the bipartite graph construction is provided in Appendix C.

Following the bipartite graph construction, we propose our Differentiable Cluster Graph Neural Network (DC-GNN) to learn the node embeddings. An illustration of DC-GNN architecture is presented in Fig. 2. Given $X_{\text{input}}$ as input node features, we first transform it with an MLP to produce $X$. The transformed features together with the bipartite graph $\mathcal{G}$ are then put through an iterative `DC-MsgPassing` algorithm to produce the embeddings of nodes $Z$ and cluster-nodes $C$. A learnable readout function such as a simple MLP, is then used on $Z$ to produce the class probabilities $Y$ in our node classification task. Then, DC-GNN is trained end-to-end with a task-specific loss function

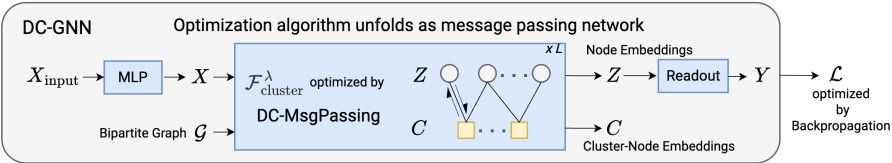

Figure 2: Overview of DC-GNN. `DC-MsgPassing` is an iterative optimization algorithm that implicitly minimizes $\mathcal{O}_{\text{cluster}}^{\lambda}$ in each step of message passing, where $\mathcal{O}_{\text{cluster}}^{\lambda}$ is an optimal transport based clustering objective function. The output of `DC-MsgPassing` are cluster-aware embeddings which can then be optimized with any task-specific loss function $\mathcal{L}$. DC-GNN is trained end-to-end in a supervised setting.

$\mathcal{L}$ via backpropagation. At the core of our design is the iterative `DC-MsgPassing` algorithm, an implicit optimization network that optimizes a clustering-based objective function $\mathcal{O}_{\text{cluster}}^{\lambda}$.

## 3.2 CLUSTERING-BASED OBJECTIVE FUNCTION $\mathcal{O}_{\text{cluster}}$

In this work, we aim to address over-squashing and heterophily by embedding a clustering inductive bias into the design of the GNN. One way to achieve this is by optimizing a clustering-based objective function. Motivated by the theory of optimal transport, we propose a novel soft clustering-based objective function.

We conceptualize the cluster assignment problem as an optimal transport problem (Villani et al., 2009), where the cost is defined as the distance between the embeddings of nodes and their cluster centroids. Therefore, we propose to minimize the overall cost, weighted by the soft cluster assignment matrix $P$ which indicates the amount of assignment from a node to a cluster. Specifically, we have a single global soft cluster assignment matrix $P^{\Omega} \in \mathbb{R}_+^{|\mathcal{V}| \times |\Omega|}$ and local soft cluster assignment matrices $P^{\Gamma_i} \in \mathbb{R}_+^{|\mathcal{N}_i^+| \times |\Gamma_i|}$ for each node $i$. Let $d(u, v)$ represent the distance between two vectors $u$ and $v$, $z_i$ be the node embeddings to be learnt for each node $i \in \mathcal{V}$, $x_i$ be the node features after an initial transformation by a multilayer perceptron, and $c_j^{\Omega}, c_j^{\Gamma_i}$ be the embeddings of the $j^{th}$ global and local cluster-node respectively. Then we define our objective function $\mathcal{O}_{\text{cluster}}$ as

$$\mathcal{O}_{\text{cluster}} = \alpha \underbrace{\sum_{i \in \mathcal{V}} \sum_{j \in \Omega} P_{ij}^{\Omega} d(z_i, c_j^{\Omega})}_{\text{global clustering}} + (1 - \alpha) \underbrace{\sum_{i \in \mathcal{V}} \sum_{u \in \mathcal{N}_i^+} \sum_{j \in \Gamma_i} P_{uj}^{\Gamma_i} d(z_u, c_j^{\Gamma_i})}_{\text{local clustering}} + \beta \underbrace{\sum_{i \in \mathcal{V}} d(z_i, x_i)}_{\text{node fidelity}}. \tag{1}$$

The *global clustering* part optimizes the OT distance between node embeddings and global cluster-node embeddings. The *local clustering* term optimizes the OT distance between the embeddings of nodes and local cluster-nodes within each ego-neighborhood. The scalar parameter $\alpha \in [0, 1]$ is a balancing factor between the two terms. Furthermore, the additional *node fidelity* term encourages the node embeddings to retain some information from the original node features (Klicpera et al., 2018; Chen et al., 2020).

## 3.3 `DC-MsgPassing` : OPTIMIZE $\mathcal{O}_{\text{cluster}}$ WITH ENTROPIC REGULARIZATION VIA MESSAGE PASSING

Since conventional OT solvers can be computationally prohibitive (Pele and Werman, 2009), we adopt the entropy regularized version of the OT distance that is designed for efficiency and offers a good approximation of OT distance (Cuturi, 2013), with details in Section 3.3.1. Let $h(P)$ be the entropy of the assignment matrix $P$, we propose the following refined objective function

$$\mathcal{O}_{\text{cluster}}^{\lambda} := \mathcal{O}_{\text{cluster}} - \frac{\alpha}{\lambda} h(P^{\Omega}) - \frac{(1 - \alpha)}{\lambda} \sum_{i \in \mathcal{V}} h(P^{\Gamma_i}). \tag{2}$$

Direct optimization of $\mathcal{O}_{\text{cluster}}^{\lambda}$ is difficult due to the presence of unobserved assignment matrices $P^{\Omega}$ and $P^{\Gamma_i}$, since we cannot compute Eq. (2) without estimating the cluster assignment values. To overcome this, we propose an iterative block coordinate descent algorithm called `DC-MsgPassing`.

In each iteration of `DC-MsgPassing`, we alternatively optimize $\mathcal{O}_{\text{cluster}}^{\lambda}$ with respect to one block of variables at a time, while all other variables are held constant. Specifically, we alternatively update the assignment matrices $P^{\Omega}$, $P^{\Gamma_i}$ and embeddings $Z, C$ in each iteration.

For the first step of solving cluster-assignments, we adopt the entropic regularized Sinkhorn distance (Cuturi, 2013) approximation for solving the OT problem. This approximation utilizes differentiable operations, allowing the clustering algorithm to be used as a component within an end-to-end learning process. For the second step of updating the node and cluster embeddings, we show that a closed-form solution can be derived to minimize the objective function given the assignment matrix.

### 3.3.1 ASSIGNMENT UPDATE

**Update for assignment matrices $P^{\Omega}$, $P^{\Gamma_i}$:**   With node and cluster-node embeddings kept constant, we aim to update the $P^{\Omega}$, $P^{\Gamma_i}$ minimizing $\mathcal{O}_{\text{cluster}}$. We approach this cluster assignment problem from the perspective of optimal transport theory (Villani et al., 2009), and use $P^{\Omega}$ as an example without loss of generality.

Our objective is to determine the optimal assignment matrix $P^{\Omega*} \in \mathbb{R}_{+}^{|\mathcal{V}| \times |\Omega|}$ for a given cost matrix $M \in \mathbb{R}_{+}^{|\mathcal{V}| \times |\Omega|}$, aligning with a target clustering distribution. Assuming a uniform distribution of data points across all clusters, we aim to find a mapping from $\mathbf{u}^{\top} = \begin{bmatrix} |\mathcal{V}|^{-1} & \cdots & |\mathcal{V}|^{-1} \end{bmatrix}_{1 \times |\mathcal{V}|}$ to $\mathbf{v}^{\top} = \begin{bmatrix} |\Omega|^{-1} & \cdots & |\Omega|^{-1} \end{bmatrix}_{1 \times |\Omega|}$, minimizing the overall cost. To formalize, optimizing $\mathcal{O}_{\text{cluster}}$ with respect to $P^{\Omega}$ is tantamount to solving:

$$\min_{P^{\Omega} \in U(\mathbf{u}, \mathbf{v})} \langle P^{\Omega}, M \rangle, \tag{3}$$

where $U(\mathbf{u}, \mathbf{v}) = \{P \in \mathbb{R}_{+}^{|\mathcal{V}| \times |\Omega|} : P\mathbf{1}_{|\Omega|} = \mathbf{u}, P^{\top}\mathbf{1}_{|\mathcal{V}|} = \mathbf{v}\}$. $\langle \cdot, \cdot \rangle$ is the Frobenius dot-product, and $\langle P^{\Omega}, M \rangle = \sum_{i \in \mathcal{V}} \sum_{j \in \Omega} P_{ij}^{\Omega} d(z_i, c_j^{\Omega})$. Importantly, $M_{ij} = d(z_i, c_j^{\Omega})$ can be seen as the cost of assigning node $i$ to cluster $j$, and $P_{ij}^{\Omega}$ indicates the amount of assignment from node $i$ to cluster $j$.

This is a classical optimal transport problem (Villani et al., 2009), where Eq. (3) represents the optimal transport distance, also known as the earth mover's distance. While such problems are typically solved via linear programming techniques, these approaches are computationally expensive (Pele and Werman, 2009). To overcome it, we opt for the Sinkhorn distance (Cuturi, 2013) instead, which offers a good approximation to the optimal transport distance with additional entropic regularization, weighted by scalar $1/\lambda$, where $\lambda > 0$. Formally,

$$\langle P_{\lambda}^{\Omega}, M \rangle, \text{ where } P_{\lambda}^{\Omega} = \underset{P^{\Omega} \in U(\mathbf{u}, \mathbf{v})}{\arg\min} \langle P^{\Omega}, M \rangle - \frac{1}{\lambda} h(P^{\Omega}). \tag{4}$$

The benefit of having this entropic regularization term $h(P^{\Omega})$ is that the solution $P_{\lambda}^{\Omega*}$ now has the form $P_{\lambda}^{\Omega*} = UBV$ (Cuturi, 2013), where $B = e^{-\lambda M}$, and $U$ and $V$ are diagonal matrices. Now the OT problem reduces to the classical matrix scaling problem (Idel, 2016), for which the objective is to determine if there exist diagonal matrices $U$ and $V$ such that the $i^{th}$ row of the matrix $UBV$ sums to $\mathbf{u}_i$ and the $j^{th}$ column of $UBV$ sums to $\mathbf{v}_j$. Since $e^{-\lambda M}$ is strictly positive, there exists a unique $P_{\lambda}^{\Omega*}$ that belongs to $U(\mathbf{u}, \mathbf{v})$ (Menon, 1968; Sinkhorn, 1967), which can be obtained by the well-known Sinkhorn–Knopp algorithm (Sinkhorn, 1967; Sinkhorn and Knopp, 1967).

To obtain $P_{\lambda}^{\Omega*}$, we run the Sinkhorn–Knopp algorithm which iteratively updates the matrix $B$ by scaling each row of $B$ by the respective row-sum, and each column of $B$ by the respective column-sum. Formally, we have

$$B_{ij}^{(t)} = \frac{B_{ij}^{(t-1)}}{\sum_j B_{ij}^{(t-1)}} \mathbf{u}_i, \quad B_{ij}^{(t+1)} = \frac{B_{ij}^{(t)}}{\sum_i B_{ij}^{(t)}} \mathbf{v}_j. \tag{5}$$

After $T$ steps, we update $P_{ij}^{\Omega}$ by the value of $B_{ij}$. Similarly, we update $P^{\Gamma_i}$ in the *local clustering* term. The scaling operations in Sinkhorn–Knopp are *fully differentiable*, enabling end-to-end learning.

### 3.3.2 EMBEDDINGS UPDATE

With the updated assignment matrices $P^{\Omega}$, $P^{\Gamma_i}$, we now derive the message passing equations to update the cluster-node and node embeddings.

**Update for cluster-node embeddings $C$:** To minimize $\mathcal{O}_{\text{cluster}}^{\lambda}$ with respect to a specific cluster node $c_j$, we differentiate $\mathcal{O}_{\text{cluster}}^{\lambda}$ in terms of $c_j$ with other variables fixed and set the derivative to zero. If we choose the distance function $d(\cdot)$ to be the squared Euclidean norm, $\mathcal{O}_{\text{cluster}}^{\lambda}$ is a quadratic model of node embeddings $c_j$. Then we can derive the following closed-form solution:

$$C = \text{diag}(\mathbf{k})P^{\top}Z, \tag{6}$$

where $\mathbf{k}^{\top} = \left[|\Omega|\mathbf{1}_{|\Omega|}; |\Gamma_i|\mathbf{1}_{|\Gamma_i|}; \cdots ; |\Gamma_{|\mathcal{V}|}|\mathbf{1}_{|\Gamma_i|}\right]_{1 \times |\mathcal{C}|}$, ; denotes concatenation. $P \in \mathbb{R}_{+}^{|\mathcal{V}| \times |\mathcal{C}|}$ is the overall assignment matrix that indicates the amount of assignment from all nodes $Z$ to all cluster-nodes $C$. See Appendix B.2 and B.3 for notation and full derivation.

*Remark* 3.1. Since $P_{ij}$ can be viewed as edge weight between a node $i \in \mathcal{V}$ and a cluster-node $j \in \mathcal{C}$, the updating mechanism serves as the message passing function from nodes to cluster-nodes.

**Update for node embeddings $Z$:** Similar to cluster-node embeddings update, we derive the node embeddings update by differentiating $\mathcal{O}_{\text{cluster}}^{\lambda}$ with respect to $z_i$. Consequently, we obtain

$$Z = \gamma\Big[\beta X + \alpha P^{\Omega}C^{\Omega} + (1-\alpha)\sum_{u \in \mathcal{N}_i^+} \hat{P}^{\Gamma_u}C^{\Gamma_u}\Big], \tag{7}$$

where $\gamma = (\alpha|\mathcal{V}|^{-1} + \beta + 1 - \alpha)^{-1}$ is a constant, and $\hat{P}^{\Gamma_u} \in \mathbb{R}_{+}^{|\mathcal{V}| \times |\Gamma_u|}$ is the broadcasted local assignment matrix from $P^{\Gamma_u} \in \mathbb{R}_{+}^{|\mathcal{N}_u^+| \times |\Gamma_u|}$ (see Appendix B.2 and B.4 for notation and full derivation). Intriguingly, this closed-form solution reveals that $Z$ is updated by a linear combination of global and local cluster-node embeddings, weighted by cluster assignment probabilities. The hyperparameter $\alpha$ balances the influence of local and long-range interactions while $\beta$ scales the original node features that provide the initial residual (Klicpera et al., 2018; Chen et al., 2020).

*Remark* 3.2. Viewing the cluster assignment probabilities as edge weights, Eq. (7) represents the message passing from cluster-nodes back to original nodes. Therefore, Eq. (6) and Eq. (7) function as message passing on the bipartite graph, enforcing the clustering inductive bias.

In summary, `DC-MsgPassing` is the key to our method. Each iteration of `DC-MsgPassing` consists of two alternative steps. First, with fixed embeddings $Z$ and $C$, optimal clustering assignment matrices are calculated via Sinkhorn-Knopp algorithm (Eq. (5)). Then, the cluster-node and node embeddings are refined through message passing with the updated assignment matrices via Eq. (6) and Eq. (7). In practice, we could also add learnable components such as linear transformation matrices or MLPs for the messages in Eq. (6) and Eq. (7) to allow the network to fit the data distribution better. *Each iteration of* `DC-MsgPassing` *is one optimization step towards minimizing* $\mathcal{O}_{\text{cluster}}^{\lambda}$. Thus our message passing mechanism provides the needed inductive bias in learning the local and global clusters present in the data, while simultaneously learning node embeddings using the clustering information. We present the details of `DC-MsgPassing` in Algorithm 1.

---

**Algorithm 1** `DC-MsgPassing`

---

**Input:** Bipartite graph $\mathcal{G} = (\mathcal{V}, \mathcal{C}, \mathcal{E})$, Node features $X$, hyperparameters $\alpha, \beta, \lambda$
**Output:** $[z_i]_{i \in \mathcal{V}}$
1: $Z = X$
2: Initialize cluster embeddings $C$
3: // Optimize $\mathcal{O}_{\text{cluster}}^{\lambda}$ via `DC-MsgPassing`
4: **for** $l = 1, 2, \ldots, L$ **do**
5:     **Update Cluster Assignment Matrices:**
6:         $M_{ij} = d(z_i, c_j) \quad \forall i \in \mathcal{V}, j \in \mathcal{C}$
7:         $B^{\{\Omega, \Gamma_i\}} = e^{-\lambda M}$
8:         // Run Sinkhorn algorithm for $T$ steps (Eq. 5)
9:         $P^{\{\Omega, \Gamma_i\}} = B^{\{\Omega, \Gamma_i\}}$
10:    **Update Node and Cluster-node Embeddings:**
11:      Calculate $C$ as per Eq. (6)
12:      Calculate $Z$ as per Eq. (7)
13: **end for**
14: **Return:** Z

---

In DC-GNN, both local and global cluster-nodes serve important but different roles. The local cluster-node connections help to preserve the original graph structure information and enable clustered aggregation. Therefore, a single update via local cluster-nodes is analogous to one layer of message passing in conventional GNNs, with the cluster structure assisting in handling heterophilic local neighbourhoods. Simultaneously, each update via global cluster-nodes allows transfer of long-range information from relevant distant nodes.

**Convergence analysis** The `DC-MsgPassing` algorithm is generally well behaved, converging when enough iterations are performed. The following theorem presents the convergence analysis of minimizing the objective function $\mathcal{O}^\lambda_{cluster}$ with the `DC-MsgPassing` algorithm.

**Theorem 3.3** (Convergence of `DC-MsgPassing`). *Assuming the Sinkhorn–Knopp algorithm is run to convergence in each iteration, for any $\lambda > 0$, the value of $\mathcal{O}^\lambda_{cluster}$ produced by* `DC-MsgPassing` *algorithm (Algorithm 1) is guaranteed to converge.*

Proof can be found in the Appendix B.1.

**Complexity analysis** The time complexity of the global assignment update step is $O(T|\mathcal{V}||\Omega|)$. In practice, it could be simplified as $O(|\mathcal{V}|)$ since $|\Omega| \ll |\mathcal{V}|$ and $T$ are small constants. Both the time complexity of the local assignment update step $O(T|\mathcal{E}|) = O(|E|)$ and the embeddings update step $O(|\mathcal{E}|) = O(|E|)$ are linear w.r.t. the number of edges. Thus the overall computational complexity is linear w.r.t. the size of the original graph $O(|E|)$, same order as standard GNNs. Runtime of `DC-MsgPassing` is measured in Appendix E.3. The memory complexity is $O(|\mathcal{V}| + |\mathcal{C}|) = O(|\mathcal{V}| + |\Omega| + \sum_{i \in \mathcal{V}} |\Gamma_i|) = O(|\Gamma_i||\mathcal{V}|)$ since $|\Omega| \ll |\mathcal{V}|$, which is linear w.r.t. the number of nodes, and $|\Gamma_i|$ the number of local clusters for each node is a small constant in practice.

### 3.4 TRAINING OF DC-GNN

In this work, we mainly focus on the supervised node classification task. As depicted in Fig.2, the node representations $Z$, generated by `DC-MsgPassing`, are subsequently passed through a readout function (e.g., a multi-layer perceptron) to produce the final output, which is used as input to a task-specific loss function $\mathcal{L}$ for end-to-end training. We use cross entropy loss $\mathcal{L}_{ce}$ to train DC-GNN along with two regularizing loss functions to facilitate the learning process.

$$\mathcal{L} = \mathcal{L}_{ce} + \omega_1 \mathcal{L}_{ortho} + \omega_2 \mathcal{L}_{sim}, \tag{8}$$

where $\mathcal{L}_{ortho}$ and $\mathcal{L}_{sim}$ are orthogonality and similarity losses, weighted by hyperparameters $\omega_1$ and $\omega_2$, as described below.

**Orthogonality loss ($\mathcal{L}_{ortho}$):** To encourage the clusters to be distinct, we adopt a regularizing orthogonality loss function (Bianchi et al., 2020) $\mathcal{L}_{ortho} = \left\| \frac{C^\top C}{\|C^\top C\|_F} - \frac{I_{|\Omega|}}{\sqrt{|\Omega|}} \right\|_F$, where $\| \cdot \|_F$ is the Frobenius norm. This pushes the cluster-nodes to be orthogonal to each other.

**Similarity loss ($\mathcal{L}_{sim}$):** To further enhance the clustering process, we introduce $\mathcal{L}_{sim}$ that encourages node similarity to only a single cluster. To achieve this, we set $|\Omega|$ to be multiple of the number of classes and associate a set of cluster-nodes $\Omega^\tau$ with each class $\tau$. We then compute distances between the node embedding and the cluster-node embeddings associated with its labelled class, select the cluster-node embedding that is most similar to the node with a $max$ operator, and push them closer. If a training node $i$ belongs to class $\tau$, $c_j^\tau$ is the $j^{th}$ cluster embedding associated with class $\tau$. Let $\Lambda$ be a similarity function, and $\mathcal{V}_+$ be the set of training nodes, then $\mathcal{L}_{sim}$ is defined as

$$\mathcal{L}_{sim} = \frac{1}{|\Omega||\mathcal{V}_+|} \sum_{i \in \mathcal{V}_+} \left[ s_i^\tau + \log \sum_{\tau' \neq \tau} \exp\left(-s_i^{\tau'}\right) \right], \text{ where } s_i^\tau = \max_{j \in |\Omega^\tau|} \Lambda\left(z_i^L, c_j^\tau\right). \tag{9}$$

## 4 EXPERIMENTS

In this section, we empirically validate the capabilities of our proposed solution through extensive experiments and ablation studies. Descriptions and statistics of the datasets are in Appendix. E.6. Baselines, implementation and training details can be found in Appendix. E.7.

### 4.1 COMPARISON WITH BASELINES ON HETEROPHILOUS GRAPHS

We first conduct experiments on heterophilous datasets where long-range information is beneficial and neighborhood aggregation needs special attention. We achieve state-of-the-art on all six heterophilous

Table 1: Classification performance comparison across various heterophilous and homophilous datasets. We report ROC AUC for Genius (Lim et al., 2021) and accuracy for other datasets. OOM refers to out-of-memory.

| | Heterophilous | | | | | | Homophilous | | |
|---|---|---|---|---|---|---|---|---|---|
| | **Penn94** | **Genius** | **Cornell5** | **Amherst41** | **US-election** | **Wisconsin** | **Cora** | **Citeseer** | **Pubmed** |
| MLP | 73.61 (0.40) | 86.68 (0.09) | 68.86 (1.83) | 60.43 (1.26) | 81.92 (1.01) | 85.29 (3.31) | 75.69 (2.00) | 74.02 (1.90) | 87.16 (0.37) |
| GCN | 82.47 (0.27) | 87.42 (0.37) | 80.15 (0.37) | 81.41 (1.70) | 82.07 (1.65) | 51.76 (3.06) | 86.98 (1.27) | 76.50 (1.36) | 88.42 (0.50) |
| GAT | 81.53 (0.55) | 55.80 (0.87) | 78.96 (1.57) | 79.33 (2.09) | 84.17 (0.98) | 49.41 (4.09) | 87.30 (1.10) | 76.55 (1.23) | 86.33 (0.48) |
| MixHop | 83.47 (0.71) | 90.58 (0.16) | 78.52 (1.22) | 76.26 (2.56) | 85.90 (1.55) | 75.88 (4.90) | 87.61 (0.85) | 76.26 (1.33) | 85.31 (0.61) |
| GCNII | 82.92 (0.59) | 90.24 (0.09) | 78.85 (0.78) | 76.02 (1.38) | 82.90 (0.29) | 80.39 (3.40) | 88.37 (1.25) | 77.33 (1.48) | 90.15 (0.43) |
| H$_2$GCN | 81.31 (0.60) | OOM | 78.46 (0.75) | 79.64 (1.63) | 85.53 (0.77) | 87.65 (4.98) | 87.87 (1.20) | 77.11 (1.57) | 89.49 (0.38) |
| WRGAT | 74.32 (0.53) | OOM | 71.11 (0.48) | 62.59 (2.46) | 84.45 (0.56) | 86.98 (3.78) | 88.20 (2.26) | 76.81 (1.89) | 88.52 (0.92) |
| GPR-GNN | 81.38 (0.16) | 90.05 (0.31) | 73.30 (1.87) | 67.00 (1.92) | 84.49 (1.09) | 82.94 (4.21) | 87.95 (1.18) | 77.13 (1.67) | 87.54 (0.38) |
| GGCN | 73.62 (0.61) | OOM | 71.35 (0.81) | 66.53 (1.61) | 84.71 (2.60) | 86.86 (3.29) | 87.95 (1.05) | 77.14 (1.45) | 89.15 (0.37) |
| ACM-GCN | 82.52 (0.96) | 80.33 (3.91) | 78.17 (1.42) | 70.11 (2.10) | 85.14 (1.33) | 88.43 (3.22) | 87.91 (0.95) | 77.32 (1.70) | 90.00 (0.52) |
| LINKX | 84.71 (0.52) | 90.77 (0.27) | 83.46 (0.61) | 81.73 (1.94) | 84.08 (0.67) | 75.49 (5.72) | 84.64 (1.13) | 73.19 (0.99) | 87.86 (0.77) |
| GloGNN++ | 85.74 (0.42) | 90.91 (0.13) | 83.96 (0.46) | 81.81 (1.50) | 85.48 (1.19) | 88.04 (3.22) | 88.33 (1.09) | 77.22 (1.78) | 89.24 (0.39) |
| DC-GNN | **86.69** (0.22) | **91.70** (0.08) | **84.68** (0.24) | **82.94** (1.59) | **89.59** (1.60) | **91.67** (1.95) | **89.13** (1.18) | **77.93** (1.82) | **91.00** (1.28) |

Table 2: Classification performance comparison on more recent heterophilous datasets (Platonov et al., 2023). We report accuracy for Roman-empire and Amazon-ratings, and ROC AUC for the rest.

| | Roman-empire | Amazon-ratings | Minesweeper | Tolokers | Questions |
|---|---|---|---|---|---|
| ResNet | 65.88 (0.38) | 45.90 (0.52) | 50.89 (1.39) | 72.95 (1.06) | 70.34 (0.76) |
| ResNet+SGC | 73.90 (0.51) | 50.66 (0.48) | 70.88 (0.90) | 80.70 (0.97) | 75.81 (0.96) |
| ResNet+adj | 52.25 (0.40) | 51.83 (0.57) | 50.42 (0.83) | 78.78 (1.11) | 75.77 (1.24) |
| GCN | 73.69 (0.74) | 48.70 (0.63) | 89.75 (0.52) | 83.64 (0.67) | 76.09 (1.27) |
| SAGE | 85.74 (0.67) | 53.63 (0.39) | 93.51 (0.57) | 82.43 (0.44) | 76.44 (0.62) |
| GAT | 80.87 (0.30) | 49.09 (0.63) | 92.01 (0.68) | 83.70 (0.47) | 77.43 (1.20) |
| GAT-sep | 88.75 (0.41) | 52.70 (0.62) | 93.91 (0.35) | 83.78 (0.43) | 76.79 (0.71) |
| GT | 86.51 (0.73) | 51.17 (0.66) | 91.85 (0.76) | 83.23 (0.64) | 77.95 (0.68) |
| GT-sep | 87.32 (0.39) | 52.18 (0.80) | 92.29 (0.47) | 82.52 (0.92) | 78.05 (0.93) |
| GPS$^{GAT+Performer}$ | 87.04 (0.58) | 49.92 (0.68) | 91.08 (0.58) | 84.38 (0.91) | 77.14 (1.49) |
| NaGphormer | 74.34 (0.77) | 51.26 (0.72) | 84.19 (0.66) | 78.32 (0.95) | - |
| Exphormer | 89.03 (0.37) | 53.51 (0.46) | 90.74 (0.53) | 83.77 (0.78) | - |
| GOAT | 71.59 (1.25) | 44.61 (0.50) | 81.09 (1.02) | 83.11 (1.04) | - |
| NeuralWalker | **92.92** (0.36) | **54.58** (0.36) | **97.82** (0.40) | **85.56** (0.74) | **78.52** (1.13) |
| H$_2$GCN | 60.11 (0.52) | 36.47 (0.23) | 89.71 (0.31) | 73.35 (1.01) | 63.59 (1.46) |
| CPGNN | 63.96 (0.62) | 39.79 (0.77) | 52.03 (5.46) | 73.36 (1.01) | 65.96 (1.95) |
| GPR-GNN | 64.85 (0.27) | 44.88 (0.34) | 86.24 (0.61) | 72.94 (0.97) | 55.48 (0.91) |
| FSGNN | 79.92 (0.56) | 52.74 (0.83) | 90.08 (0.70) | 82.76 (0.61) | 78.86 (0.92) |
| GloGNN | 59.63 (0.69) | 36.89 (0.14) | 51.08 (1.23) | 73.39 (1.17) | 65.74 (1.19) |
| FAGCN | 65.22 (0.56) | 44.12 (0.30) | 88.17 (0.73) | 77.75 (1.05) | 77.24 (1.26) |
| GBK-GNN | 74.57 (0.47) | 45.98 (0.71) | 90.85 (0.58) | 81.01 (0.67) | 74.47 (0.86) |
| JacobiConv | 71.14 (0.42) | 43.55 (0.48) | 89.66 (0.40) | 68.66 (0.65) | 73.88 (1.16) |
| GMN | 87.69 (0.50) | **54.07** (0.31) | 91.01 (0.23) | 84.52 (0.21) | - |
| Diag-NSD | 77.50 (0.67) | 37.96 (0.20) | 89.59 (0.61) | 79.81 (0.99) | 69.25 (1.15) |
| ACMP | 71.27 (0.59) | 44.76 (0.52) | 76.15 (1.12) | 75.03 (0.92) | 71.18 (1.03) |
| CDE-GRAND | **91.64** (0.28) | 47.63 (0.43) | 95.50 (5.23) | 80.70 (1.04) | 75.17 (0.99) |
| CDE-GraphBel | 85.39 (0.46) | 45.22 (0.60) | 93.98 (0.57) | 81.30 (0.43) | 72.11 (1.31) |
| DC-GNN | 89.96 (0.35) | 51.11 (0.47) | **98.50** (0.21) | **85.88** (0.81) | **78.96** (0.60) |

datasets in Tab. 1. The performance uplift is especially pronounced on US-election and Wisconsin, where our method outperforms baselines by more than 3%. Furthermore, our method achieves state-of-the-art on four out of five heterophilous datasets proposed by (Platonov et al., 2023), as shown in Tab. 2. Our performance is especially strong on Minesweeper, where we surpass existing baselines by 4%. Baseline results are from (Li et al., 2022), (Platonov et al., 2023) and (Müller et al., 2023) except for Cornell5, Amherst41 and US-election, which we reproduced following the code in (Li et al., 2022). Details are in Appendix. E.7.

## 4.2 COMPARISON WITH BASELINES ON HOMOPHILOUS GRAPHS

For a more comprehensive evaluation, we run DC-GNN on well-known homophilous citation network datasets Cora, Citeseer and Pubmed (Pei et al., 2020). As shown in Tab. 1, DC-GNN achieves the best performance on all three datasets, outperforming both general-purpose GNN baselines and those proposed specifically for heterophilous graphs. Our strong performance on both homophilous and heterophilous graphs shows that our network with the built-in clustering inductive bias is flexible and adaptive, capable of effective information aggregation in graphs with various homophily levels.

Table 3: Classification accuracy with 5 labels per class on three homophilous datasets.

|  | Cora | Citeseer | Pubmed |
|---|---|---|---|
| GCN | 69.23 (3.39) | **63.03** (4.48) | 68.00 (3.75) |
| GAT | 68.17 (5.54) | 55.54 (1.82) | 64.24 (4.79) |
| SAGE | 64.47 (1.36) | 57.10 (2.20) | 66.23 (2.65) |
| GCNII | 60.03 (7.17) | 40.40 (4.68) | 69.40 (6.01) |
| SGC | 67.80 (2.20) | 55.37 (1.16) | 63.70 (5.92) |
| Graph U-Net | 64.42 (5.44) | 49.43 (5.81) | 65.05 (4.69) |
| GraphMix | 71.99 (6.46) | 58.55 (2.26) | 67.66 (3.90) |
| MixHop | 65.33 (0.94) | 52.03 (3.92) | 71.60 (0.54) |
| DC-GNN | **72.17** (1.76) | 62.14 (2.65) | **75.07** (0.82) |

Table 4: Classification performance comparison on heterophilous datasets with 5% of original training data.

|  | Penn94 | Cornell5 | Amherst41 | US-election |
|---|---|---|---|---|
| MLP | 67.10 (1.34) | 63.33 (0.79) | 55.21 (0.84) | 79.42 (0.42) |
| GCN | 70.39 (0.84) | 69.23 (1.16) | 63.87 (1.31) | 80.55 (0.97) |
| GAT | 68.83 (1.93) | 68.04 (1.49) | 61.98 (1.59) | 81.45 (0.81) |
| MixHop | 68.37 (1.14) | 66.25 (0.67) | 61.02 (0.71) | 81.81 (0.68) |
| GCNII | 68.04 (0.18) | 66.34 (0.75) | 62.59 (2.55) | 82.28 (0.63) |
| GPR-GNN | 68.83 (0.35) | 66.69 (0.67) | 57.41 (2.00) | 81.64 (0.96) |
| GGCN | OOM | 65.23 (0.36) | 56.45 (2.29) | 79.50 (1.05) |
| ACM-GCN | 70.58 (0.43) | 65.69 (0.45) | 56.4 (1.70) | 81.60 (0.49) |
| LINKX | 69.29 (0.27) | 69.09 (0.26) | 63.27 (2.67) | 77.02 (1.43) |
| GloGNN++ | 71.29 (0.54) | 69.85 (1.02) | 63.94 (1.97) | 80.42 (0.99) |
| DC-GNN | **75.38** (0.19) | **72.47** (0.40) | **64.53** (0.19) | **83.82** (0.10) |

### 4.3 BENEFIT OF CAPTURING LONG RANGE INFORMATION IN SPARSE LABEL SETTINGS

Our method introduces shortcuts between distant nodes via the cluster-nodes. To empirically evaluate the effects of shortcut construction, we consider a generalized scenario on homophilous graphs where information from labeled nodes needs to propagate over a long distance to reach most unlabeled nodes. Specifically, we evaluate our method with only a small number of training labels per class.

Our hypothesis is that information propagation becomes more challenging when useful training information is more scarce, making the effects of shortcut construction more pronounced in sparsely-annotated graph datasets. The hypothesis is supported by experiment results in Tab. 3, where our method outperforms other methods on Cora and Pubmed by substantial margins. We have also conducted low label rate experiments on heterophilous datasets, with just 5% of training labels. As shown in Tab. 4, DC-GNN continues to outperform the baselines across four heterophilous datasets, with a notable 4% improvement on Penn94.

### 4.4 ALLEVIATING OVERSQUASHING

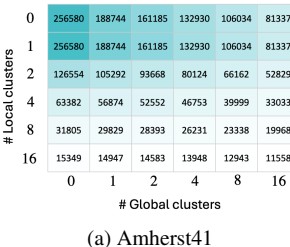

(a) Amherst41

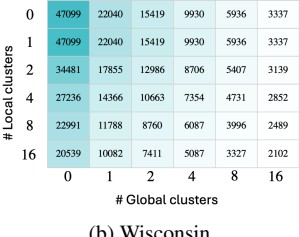

(b) Wisconsin

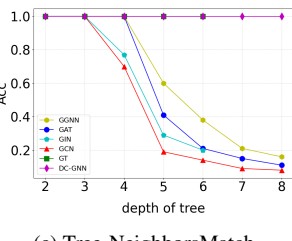

(c) Tree-NeighborsMatch

Figure 3: (a)-(b) Total effective resistance heatmap. (c) Accuracy for Tree-NeighborsMatch dataset.

Total effective resistance ($R_{\text{tot}}$) is established as an indicator of oversquashing (Black et al., 2023), prompting the development of various graph rewiring strategies to diminish $R_{\text{tot}}$ within the underlying graph and thus address oversquashing. Our approach contributes to this endeavor by introducing cluster-nodes. This effectively creates new pathways among the original nodes, thereby reducing the graph's $R_{\text{tot}}$ and aiding in mitigating the oversquashing issue (Black et al., 2023). To validate this, we conduct an empirical analysis of the total pairwise effective resistance among the original nodes in our bipartite graph, with varying number of global and local clusters. Fig. 3a and Fig. 3b display a heatmap of $R_{\text{tot}}$, with darker shades representing higher $R_{\text{tot}}$ values. The results indicate that $R_{\text{tot}}$ decreases sharply as we increase the number of both global and local clusters. In addition, we also perform experiments on synthetic random graphs with varying degrees of sparsity and show that the bipartite graph construction reduces $R_{\text{tot}}$ on these graphs. Details can be found in Appendix E.1.

To further validate the oversquashing mitigation capability, we conduct experiments on Tree-NeighborsMatch dataset (Alon and Yahav, 2020), which requires long-range interaction between leaf nodes and the root node of tree graphs with varying depths. In Fig. 3c, DC-GNN achieves perfect performance along with GT (Müller et al., 2023) on all depth settings, significantly outperforming other message passing GNNs.

## 4.5 ABLATION STUDIES

### 4.5.1 EFFECTS OF EACH TERM IN $\mathcal{O}_{\text{cluster}}^{\lambda}$

To validate the effectiveness of our `DC-MsgPassing` algorithm, we conduct an ablation study on the individual components of our objective function $\mathcal{O}_{\text{cluster}}^{\lambda}$. Specifically, we vary the parameters by setting (1) $\alpha$ to 0, (2) $\alpha$ to 1 and (3) $\beta$ to 0, aiming to ablate the contributions of the global clustering term, local clustering term and the node fidelity term respectively.

Results from Tab. 5 indicate that the contributions of global and local clustering vary on different datasets. Specifically, the contribution of local clustering is dominant on Genius, US-election and Amherst41. This is expected as local clustering facilitates message passing via adjacent nodes

Table 5: Effects of each term in $\mathcal{O}_{\text{cluster}}^{\lambda}$.

|  | GENIUS | US-ELECTION | PENN94 | AMHERST41 |
|---|---|---|---|---|
| DC-GNN | 91.70 (0.08) | 89.59 (1.60) | 86.69 (0.22) | 82.94 (1.59) |
| (-)GLOBAL | 91.62 (0.07) | 88.77 (2.21) | 84.61 (0.42) | 81.43 (1.53) |
| (-)LOCAL | 87.05 (0.09) | 83.26 (1.77) | 86.69 (0.22) | 80.77 (2.04) |
| (-)FIDELITY | 91.08 (0.04) | 87.84 (2.67) | 86.69 (0.22) | 82.28 (1.32) |

and embeds graph structure information into the model. The contribution of global clustering is most pronounced on Penn94, indicating the usefulness of long-range information in Penn94 captured by global clustering term. Additionally, the results show that all three terms—local clustering, global clustering, and node fidelity—contribute to the overall efficacy of our model.

### 4.5.2 EFFECTS OF $\mathcal{L}_{\text{ortho}}$ AND $\mathcal{L}_{\text{sim}}$

We introduced *orthogonality* ($\mathcal{L}_{\text{ortho}}$) and *similarity* ($\mathcal{L}_{\text{sim}}$) losses to regularize and assist the clustering process. In this ablation, we evaluate the effects of these losses on model performance. As shown in Tab. 6, $\mathcal{L}_{\text{ortho}}$ and $\mathcal{L}_{\text{sim}}$ generally help to improve the scores across datasets. While

Table 6: Effects of $\mathcal{L}_{\text{ortho}}$ and $\mathcal{L}_{\text{sim}}$.

|  | GENIUS | US-ELECTION | PENN94 | AMHERST41 |
|---|---|---|---|---|
| DC-GNN | 91.70 (0.08) | 89.59 (1.60) | 86.69 (0.22) | 82.94 (1.59) |
| (-) $\mathcal{L}_{\text{sim}}$ | 91.70 (0.08) | 89.08 (1.46) | 86.65 (0.15) | 82.22 (1.03) |
| (-) $\mathcal{L}_{\text{ortho}}$ | 91.68 (0.08) | 88.72 (1.20) | 86.64 (0.27) | 82.35 (1.25) |
| (-)$\mathcal{L}_{\text{sim}}, \mathcal{L}_{\text{ortho}}$ | 91.68 (0.08) | 88.61 (1.57) | 86.40 (0.25) | 81.75 (1.07) |

$\mathcal{O}_{\text{cluster}}^{\lambda}$ is central to our model, the auxiliary losses $\mathcal{L}_{\text{ortho}}$ and $\mathcal{L}_{\text{sim}}$ play a more supportive role to facilitate the clustering process, likely by promoting distinct cluster representations and enhancing node-cluster alignment, as hypothesized. The modest performance uplift suggests that these losses make a positive, albeit limited, contribution, underscoring the dominant influence of $\mathcal{O}_{\text{cluster}}^{\lambda}$ in DC-GNN's superior performance. More results can be found in App. E.2.2.

## 5 CONCLUSION AND FUTURE WORK

This paper tackles the dual challenges in Graph Neural Networks: capturing global long-range information and preserving performance in heterophilous local neighborhoods. We proposed a novel differentiable framework that seamlessly embeds a clustering inductive bias into the message passing mechanism, facilitated by the introduction of cluster-nodes. At the heart of our approach is an optimal transport based implicit clustering objective function whose optimization presents a considerable challenge. We addressed this through an iterative optimization strategy, alternating between the computation of cluster assignments and the refinement of node/cluster-node embeddings. Importantly, the derived optimization steps effectively function as message passing steps on the bipartite graph. The message passing algorithm is efficient and we show that it is guaranteed to converge. The efficacy of our clustering-centric method in capturing both local nuances and global structures within graphs is supported by extensive experiments on both heterophilous and homophilous datasets.

Finally, we identify some limitations and future directions of our approach. Firstly, our work is motivated by the node classification task. The alignment of our proposed clustering inductive bias with graph-level tasks is somewhat unclear. We conduct some preliminary experiments with mixed results (Appendix E.4). To better align with graph-level tasks, one possible direction is to share clusters across different graphs. Secondly, the idea of embedding clustering inductive bias could potentially be extended to hierarchical clustering, which might be beneficial on very large graphs.

## REPRODUCIBILITY STATEMENT

Details of the datasets used can be found in Appendix E.6. For experiments, we document implementation details in Appendix E.7, training settings in Appendix E.7 and hyperparameter details in Appendix E.7.

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

## A    NOTATIONS

All notations are listed in Tab. 7.

Table 7: Table for notations.

| Variable | Definition |
| --- | --- |
| $\mathcal{G}$ | bipartite graph, denoted as $(\mathcal{V}, \mathcal{C}, \mathcal{E})$ |
| $G$ | original graph, denoted as $(V, E)$ |
| $V$ | set of nodes from the original graph $G$ |
| $\mathcal{V}$ | vertices of $\mathcal{G}$, direct copy of $V$ |
| $E$ | set of edges from the original graph $G$ |
| $\mathcal{E}$ | set of edges in the bipartite graph $\mathcal{G}$ |
| $\mathcal{N}_i$ | set of one-hop neighbors of node $i$ |
| $\mathcal{N}_i^+$ | node $i$ and its one-hop neighbors (ego-neighborhood of $i$) |
| $\mathcal{C}$ | set of cluster-nodes in the bipartite graph $\mathcal{G}$ |
| $\Omega$ | set of global cluster-nodes |
| $\Gamma$ | set of local cluster-nodes |
| $\Gamma_i$ | set of local cluster-nodes associated with $\mathcal{N}_i^+$ |
| $C$ | set of cluster-node embeddings |
| $Z$ | set of node embeddings |
| $Y$ | predicted class probabilities |
| $X_{\text{input}}$ | input features |
| $X$ | transformed input features |
| $P$ | overall cluster assignment matrix. $P \in \mathbb{R}_+^{|\mathcal{V}| \times |\mathcal{C}|}$ |
| $P^\Omega$ | global cluster assignment matrix. $P^\Omega \in \mathbb{R}_+^{|\mathcal{V}| \times |\Omega|}$ |
| $P^\Gamma$ | local cluster assignment matrix. $P^{\Gamma_i} \in \mathbb{R}_+^{|\mathcal{N}_i^+| \times |\Gamma_i|}$ for each node $i$ |
| $d(\cdot)$ | distance function |
| $z_i$ | node embeddings of node $i$ |
| $x_i$ | node features of node $i$ after initial transformation |
| $c_j^\Omega$ | node embeddings of $j^{th}$ global cluster-node |
| $c_j^{\Gamma_i}$ | node embeddings of $j^{th}$ local cluster-node for $\mathcal{N}_i^+$ |
| $\alpha$ | scalar parameter that balances global and local clustering objectives. $\alpha \in [0, 1]$ |
| $\beta$ | scalar parameter for node fidelity term |
| $h(\cdot)$ | entropy function |
| $P^{\Omega*}$ | optimal global soft-assignment matrix. $P^{\Omega*} \in \mathbb{R}_+^{|\mathcal{V}| \times |\Omega|}$ |
| $M_{ij}$ | cost of assigning node $i$ to cluster $j$ |
| $M$ | cost matrix |
| $\mathbf{u}^\top$ | $\mathbf{u}^\top = \begin{bmatrix} |\mathcal{V}|^{-1} & \cdots & |\mathcal{V}|^{-1} \end{bmatrix}_{1 \times |\mathcal{V}|}$ |
| $\mathbf{v}^\top$ | $\mathbf{v}^\top = \begin{bmatrix} |\Omega|^{-1} & \cdots & |\Omega|^{-1} \end{bmatrix}_{1 \times |\Omega|}$ |
| $U(\mathbf{u}, \mathbf{v})$ | $U(\mathbf{u}, \mathbf{v}) = \{P \in \mathbb{R}_+^{|\mathcal{V}| \times |\Omega|} : P\mathbf{1}_{|\Omega|} = \mathbf{u}, P^\top \mathbf{1}_{|\mathcal{V}|} = \mathbf{v}\}$ |
| $\langle \cdot, \cdot \rangle$ | Frobenius dot product |
| $P_{ij}^\Omega$ | the amount of assignment from node $i \in \mathcal{V}$ to cluster $j \in \Omega$ |
| $\lambda$ | scalar for entropy regularization |
| $B$ | initial value $e^{-\lambda M}$ |
| $U$ | diagonal matrix |
| $V$ | diagonal matrix |
| $T$ | number of iterations for Sinkhorn-Knopp algorithm |
| $\mathbf{k}$ | $\mathbf{k}^\top = \begin{bmatrix} |\Omega|\mathbf{1}_{|\Omega|}; |\Gamma_i|\mathbf{1}_{|\Gamma_i|}; \cdots ; |\Gamma_{|\mathcal{V}|}|\mathbf{1}_{|\Gamma_i|} \end{bmatrix}_{1 \times |\mathcal{C}|}$ |
| $\tau$ | class $\tau$ for our node classification task |
| $\tau'$ | class that is not class $\tau$ |
| $\Omega^\tau$ | set of global cluster-nodes associated with class $\tau$ |
| $L$ | number of `DC-MsgPassing` iterations |
| $c_j^\tau$ | the $j^{th}$ cluster embedding associated with class $\tau$ |
| $z_i^L$ | embeddings for node $i$ after $L$ iterations |
| $\gamma$ | a constant. $\gamma = (\alpha N^{-1} + \beta + 1 - \alpha)^{-1}$ |
| $\hat{P}^{\Gamma_u}$ | $\hat{P}^{\Gamma_u} \in \mathbb{R}_+^{|\mathcal{V}| \times |\Gamma_u|}$ broadcasted local cluster assignment matrix from $P^{\Gamma_u} \in \mathbb{R}_+^{|\mathcal{N}_i^+| \times |\Gamma_i|}$, defined in Eq.(16) |
| $\| \cdot \|_F$ | Frobenius norm |
| $\Lambda(\cdot)$ | similarity function |
| $\mathcal{V}_+$ | set of training nodes |

# B  PROOF AND DERIVATION

## B.1  PROOF OF THEOREM 3.3

To prove Theorem 3.3, we first introduce the following Lemma.

**Lemma B.1.** *Let* $\mathcal{O}_{\text{cluster}}^{\lambda}$ *be the objective function optimized by the* `DC-MsgPassing` *algorithm when the optimal transport components are replaced by the entropic regularized versions. Then,* $\mathcal{O}_{\text{cluster}}^{\lambda}$ *is lower-bounded by:*

$$\mathcal{O}_{\text{cluster}}^{\lambda} \geq \frac{\alpha}{\lambda} \log \frac{1}{|\mathcal{V}||\Omega|} + \frac{1-\alpha}{\lambda} \sum_{i \in \mathcal{V}} \log \frac{1}{|\mathcal{N}_i^+||\Gamma_i|}.$$

*Proof.* Recall that $\Omega$ is the set of global cluster-nodes, and $\Gamma_i$ refers to the set of local cluster-nodes associated with a node $i$. $\mathcal{V}$ is a direct copy of the nodes $V$ from the original graph. $P^{\Omega} \in \mathbb{R}^{|\mathcal{V}| \times |\Omega|}$ and $P^{\Gamma_i} \in \mathbb{R}^{|\mathcal{N}_i^+| \times |\Gamma_i|}$ are the global and local soft cluster assignment matrices respectively. $\mathcal{N}_i^+$ refers to the ego neighborhood of node $i$.

Let $p(P^{\Omega})$ be the probability of assignment matrix $P^{\Omega}$ and $h(P^{\Omega})$ be its entropy, we can upper bound its entropy by:

$$h(P^{\Omega}) = \mathbb{E}\left[\log \frac{1}{p(P^{\Omega})}\right]$$

$$\leq \log \mathbb{E}\left[\frac{1}{p(P^{\Omega})}\right] \qquad \text{(by Jensen's inequality)}$$

$$= \log \sum_{i \in \mathcal{V}, j \in \Omega} p(P_{ij}^{\Omega}) \frac{1}{p(P_{ij}^{\Omega})} \tag{10}$$

$$= \log \left(|\mathcal{V}| \times |\Omega|\right). \tag{11}$$

The inequality is due to uniform distribution having the maximum entropy.

Similarly, for local assignment matrices $P^{\Gamma_i}$, we have:

$$h(P^{\Gamma_i}) \leq \log \left(|\mathcal{N}_i^+| \times |\Gamma_i|\right). \tag{12}$$

Note that distance $d(\cdot, \cdot) \geq 0$, with the above results we obtain:

$$\mathcal{O}_{\text{cluster}}^{\lambda} = \alpha \left( \sum_{i \in \mathcal{V}} \sum_{j \in \Omega} P_{ij}^{\Omega} d(z_i, c_j^{\Omega}) - \frac{1}{\lambda} h(P^{\Omega}) \right) + \beta \sum_{i \in \mathcal{V}} d(z_i, x_i) \tag{13}$$

$$+ (1-\alpha) \sum_{i \in \mathcal{V}} \left( \sum_{k \in \mathcal{N}_i^+} \sum_{j \in \Gamma_i} P_{kj}^{\Gamma_i} d(z_k, c_j^{\Gamma_i}) - \frac{1}{\lambda} h(P^{\Gamma_i}) \right) \tag{14}$$

$$\geq -\frac{\alpha}{\lambda} h(P^{\Omega}) - \frac{(1-\alpha)}{\lambda} \sum_{i \in \mathcal{V}} h(P^{\Gamma_i}) \qquad \text{(by } d(\cdot, \cdot) \geq 0\text{)}$$

$$\geq -\frac{\alpha}{\lambda} \log \left(|\mathcal{V}| \times |\Omega|\right) - \frac{(1-\alpha)}{\lambda} \sum_{i \in \mathcal{V}} \log \left(|\mathcal{N}_i^+| \times |\Gamma_i|\right)$$

$$\text{(by equation 11 and equation 12)}$$

$$= \frac{\alpha}{\lambda} \log \frac{1}{|\mathcal{V}| \times |\Omega|} + \frac{(1-\alpha)}{\lambda} \sum_{i \in \mathcal{V}} \log \frac{1}{|\mathcal{N}_i^+| \times |\Gamma_i|}, \tag{15}$$

which concludes the proof. $\qquad\qquad\square$

By Lemma B.1, there exists a lower bound of $\mathcal{L}_{cluster}^{\lambda}$. Therefore, to prove the convergence of our algorithm, we only need to show that the loss function is guaranteed to decrease monotonically in each iteration until convergence for the assignment update step and for the embeddings update step.

For the assignment update step, let $P^{\Omega}$ be the current assignment from the previous iteration and $P^{\Omega*}$ be the new assignment obtained as

$$P^{\Omega*} \in \underset{P^{\Omega} \in U(\mathbf{u},\mathbf{v})}{\arg\min} \langle P^{\Omega*}, M \rangle - \frac{1}{\lambda} h(P^{\Omega*}).$$

The change in the loss function after this assignment step is then given by

$$\mathcal{O}_{\text{cluster}}^{\lambda}(P^{\Omega*}) - \mathcal{O}_{\text{cluster}}^{\lambda}(P^{\Omega}) \leq 0,$$

where the inequality holds by the convergence of Sinkhorn–Knopp algorithm (Sinkhorn and Knopp, 1967; Sinkhorn, 1967; Franklin and Lorenz, 1989; Soules, 1991; Chakrabarty and Khanna, 2021). Similarly, we have

$$\mathcal{O}_{\text{cluster}}^{\lambda}(P^{\Gamma_i*}) - \mathcal{O}_{\text{cluster}}^{\lambda}(P^{\Gamma_i}) \leq 0.$$

For the embeddings update step, let $Z$ and $C$ be the current embeddings from the previous iteration, and $Z^*$ and $C^*$ be the new embeddings obtained by Eq. (6) and Eq. (7). Let the $d(\cdot, \cdot)$ be the squared Euclidean distance, then since $P$ is a positive constant in this step, the loss function is convex. Since Eq. (6) and Eq. (7) are closed form solutions to the loss function, we have

$$\mathcal{O}_{\text{cluster}}^{\lambda}(Z^*, C^*) - \mathcal{O}_{\text{cluster}}^{\lambda}(Z, C) \leq 0.$$

Since $\mathcal{L}_{cluster}^{\lambda}$ has a lower bound and it decreases monotonically in each iteration, the value of $\mathcal{L}_{cluster}^{\lambda}$ produced by the message passing algorithm is guaranteed to converge.

## B.2 BROADCASTED ASSIGNMENT MATRICES

Let $f: \mathcal{Z} \times \mathcal{Z} \to \mathcal{Z}$ be a mapping from the index $k$ of a node in the ego-neighborhood of the node $u$ to the index $i$ of the same node in the node set $\mathcal{V}$. Let $P^{\Gamma_u} \in \mathbb{R}_+^{|\mathcal{N}_u^+| \times |\Gamma_u|}$ be the local assignment matrix for the ego-neighborhood of node $u \in \mathcal{V}$. For any $u \in \mathcal{V}$, we define the broadcasted local assignment matrix $\hat{P}^{\Gamma_u} \in \mathbb{R}_+^{|\mathcal{V}| \times |\Gamma_u|}$ as

$$\hat{P}_{ij}^{\Gamma_u} = \begin{cases} P_{kj}^{\Gamma_u}, & \text{if } i = f(u,k) \\ 0, & \text{otherwise} \end{cases}. \tag{16}$$

Then, we can define the the overall assignment matrix $P \in \mathbb{R}_+^{|\mathcal{V}| \times |\mathcal{C}|}$, where $|\mathcal{C}| = |\Omega| + |\Gamma|$ and $|\Gamma| = \sum_{i \in \mathcal{V}} |\Gamma_i|$, as

$$P = \begin{bmatrix} P^{\Omega} & \hat{P}^{\Gamma_1} & \cdots & \hat{P}^{\Gamma_{|\mathcal{V}|}} \end{bmatrix}. \tag{17}$$

Note that each element $P_{ij}$ can be viewed as the edge weights of a node $i \in \mathcal{V}$ and a specific cluster-node $j \in \mathcal{C}$. In simple words, P includes all the global and local cluster assignment matrices, collated in a single matrix. This allows us to unify the message passing update in Eq.( 7) for both local and global clustering terms.

## B.3 UPDATE FOR CLUSTER-NODE EMBEDDINGS $C$: DERIVATION OF EQ. (6)

Without loss of generality, we provide the full derivation for global cluster-node embeddings update function. With squared Euclidean norm as the distance function, *i.e.*, $d(u,v) = \|u - v\|^2$, we can derive that

$$\frac{\partial \mathcal{O}_{\text{cluster}}^{\lambda}}{\partial c_j^{\Omega}} = \frac{\partial}{\partial c_j^{\Omega}} \sum_{i \in \mathcal{V}} P_{ij}^{\Omega} \|z_i - c_j^{\Omega}\|^2 = 0$$

Then we have

$$2 \sum_{i \in \mathcal{V}} P_{ij}^{\Omega}(c_j^{\Omega} - z_i) = 0$$

By rearranging the terms

$$\sum_{i \in \mathcal{V}} P_{ij}^{\Omega} c_j^{\Omega} = \sum_{i \in \mathcal{V}} P_{ij}^{\Omega} z_i$$

Then one has

$$c_j^{\Omega} = \frac{\sum_{i \in \mathcal{V}} P_{ij}^{\Omega} z_i}{\sum_{i \in \mathcal{V}} P_{ij}^{\Omega}}$$

Since $P^{\Omega} \in U(\mathbf{u}, \mathbf{v})$, we have $\sum_i P_{ij}^{\Omega} = \frac{1}{|\Omega|}$ where $|\Omega|$ is the number of global clusters. Therefore,

$$c_j^{\Omega} = |\Omega| \sum_{i \in \mathcal{V}} P_{ij}^{\Omega} z_i \tag{18}$$

Similarly, with $|\Gamma_i|$ denoting the number of local clusters within the ego-neighborhood of node $i$, we have

$$c_j^{\Gamma_i} = |\Gamma_i| \sum_{u \in \mathcal{N}_i^+} P_{ij}^{\Gamma_i} z_u \tag{19}$$

Let $\mathbf{k}^{\top} = \left[|\Omega|\mathbf{1}_{|\Omega|}; |\Gamma_i|\mathbf{1}_{|\Gamma_i|}; \cdots ; |\Gamma_{|\mathcal{V}|}|\mathbf{1}_{|\Gamma_i|}\right]_{1 \times |\mathcal{C}|}$, where ; denotes concatenation. Then we can combine Eq. (18) and Eq. (19) together in one matrix equation,

$$C = \mathrm{diag}(\mathbf{k}) P^{\top} Z, \tag{20}$$

where the overall assignment matrix $P$ is defined in Eq. (17),

### B.4 UPDATE FOR NODE EMBEDDINGS $Z$: DERIVATION OF EQ. (7)

We provide the full derivation of the node embeddings update function. This manifests as message passing from cluster-nodes to nodes. With squared Euclidean norm as the distance function, *i.e.*, $d(u, v) = \|u - v\|^2$, we can derive that

$$\frac{\partial \mathcal{O}_{\mathrm{cluster}}^{\lambda}}{\partial z_i} = \alpha \sum_{j \in \Omega} 2P_{ij}^{\Omega}(z_i - c_j) + 2\beta(z_i - x_i) + (1 - \alpha) \sum_{u \in \mathcal{N}_i^+} \sum_{j \in \Gamma_u} 2P_{ij}^{\Gamma_u}(z_i - c_j) = 0 \tag{21}$$

From Eq. (21), we obtain

$$\alpha \sum_{j \in \Omega} P_{ij}^{\Omega} z_i - \alpha \sum_{j \in \Omega} P_{ij}^{\Omega} c_j + \beta z_i - \beta x_i + (1 - \alpha) \sum_{u \in \mathcal{N}_i^+} \sum_{j \in \Gamma_u} P_{ij}^{\Gamma_u} z_i - (1 - \alpha) \sum_{u \in \mathcal{N}_i^+} \sum_{j \in \Gamma_u} P_{ij}^{\Gamma_u} c_j = 0$$

By rearranging the terms, we have

$$(\alpha \sum_{j \in \Omega} P_{ij}^{\Omega} + \beta + (1 - \alpha) \sum_{u \in \mathcal{N}_i^+} \sum_{j \in \Gamma_u} P_{ij}^{\Gamma_u}) z_i = \alpha \sum_{j \in \Omega} P_{ij}^{\Omega} c_j + \beta x_i + (1 - \alpha) \sum_{u \in \mathcal{N}_i^+} \sum_{j \in \Gamma_u} P_{ij}^{\Gamma_u} c_j$$

Since $P^{\Omega} \in U(\mathbf{u}, \mathbf{v})$, we have $\sum_j P_{ij}^{\Omega} = \frac{1}{|\mathcal{V}|}$. Similarly, $\sum_{j \in \Gamma_u} P_{ij}^{\Gamma_u} = \frac{1}{|\mathcal{N}_i^+|}$. Therefore, we can deduce that

$$(\frac{\alpha}{|\mathcal{V}|} + \beta + (1 - \alpha) \sum_{u \in \mathcal{N}_i^+} \frac{1}{|\mathcal{N}_i^+|}) z_i = \alpha \sum_{j \in \Omega} P_{ij}^{\Omega} c_j + \beta x_i + (1 - \alpha) \sum_{u \in \mathcal{N}_i^+} \sum_{j \in \Gamma_u} P_{ij}^{\Gamma_u} c_j$$

With $\sum_{u \in \mathcal{N}_i^+} \frac{1}{|\mathcal{N}_i^+|} = 1$, we have

$$(\frac{\alpha}{|\mathcal{V}|} + \beta + 1 - \alpha)z_i = \alpha \sum_{j \in \Omega} P_{ij}^{\Omega} c_j + \beta x_i + (1 - \alpha) \sum_{u \in \mathcal{N}_i^+} \sum_{j \in \Gamma_u} P_{ij}^{\Gamma_u} c_j$$

This leads to

$$z_i = \frac{1}{\frac{\alpha}{|\mathcal{V}|} + \beta + 1 - \alpha}[\alpha \sum_{j \in \Omega} P_{ij}^{\Omega} c_j + \beta x_i + (1 - \alpha) \sum_{u \in \mathcal{N}_i^+} \sum_{j \in \Gamma_u} P_{ij}^{\Gamma_u} c_j]$$

Finally, expressing in matrix form admits the following closed-form solution

$$Z = \frac{1}{\frac{\alpha}{|\mathcal{V}|} + \beta + 1 - \alpha}[\alpha P^{\Omega} C^{\Omega} + \beta X + (1 - \alpha) \sum_{u \in \mathcal{N}_i^+} \hat{P}^{\Gamma_u} C^{\Gamma_u}]$$

where $\hat{P}^{\Gamma_u} \in \mathbb{R}_+^{|\mathcal{V}| \times |\Gamma_u|}$ is the broadcasted local cluster assignment matrix from $\hat{P}^{\Gamma_u} \in \mathbb{R}_+^{|\mathcal{N}_u^+| \times |\Gamma_u|}$ as defined in Eq.(16).

## C  ILLUSTRATION OF BIPARTITE GRAPH FORMULATION

We construct a bipartite graph, denoted as $\mathcal{G} = (\mathcal{V}, \mathcal{C}, \mathcal{E})$. The graph is derived from the original graph $G = (V, E)$ and comprises two distinct sets of nodes. The first set, $\mathcal{V}$, is a direct copy of the nodes $V$ from the original graph. The second set, $\mathcal{C}$, consists of cluster nodes divided into two categories: global clusters ($\Omega$) and local clusters ($\Gamma$).

In this bipartite graph, each node from the global clusters $\Omega$ connects to all nodes in $\mathcal{V}$. Meanwhile, each node from the local clusters $\Gamma$ is associated with a specific node $i$ in $\mathcal{V}$, and connected to its ego-neighborhood, which includes node $i$ and its one-hop neighbors. For a node $i$ in $\mathcal{V}$, $\Gamma_i$ represents the set of local clusters associated with it. The total number of nodes in $\mathcal{C}$ is the sum of nodes in $\Omega$ and the nodes in all local clusters $i.e.$, $|\mathcal{C}| = |\Omega| + \sum_{i \in \mathcal{V}} |\Gamma_i|$. An illustration of the bipartite graph is provided in Fig. 4.

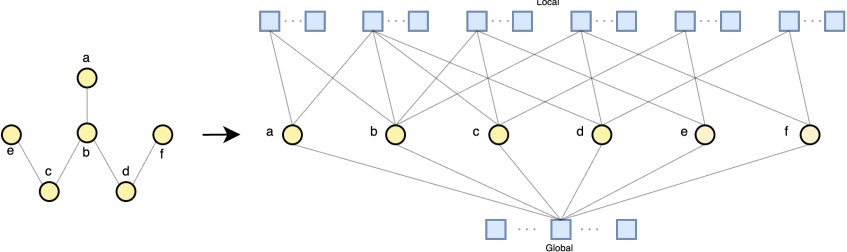

Figure 4: Based on the original graph on the left, we construct a bipartite graph on the right by adding local and global cluster-nodes. For each node in the original graph, a set of local cluster-nodes, represented by the blue boxes at the top, is connected to its ego-neighborhood. For example, the ego-neighborhood of node a includes itself and its one-hop neighbor node b. Therefore the local cluster-nodes for node a are connected to a and b. Meanwhile, a set of global cluster-nodes are added and connected to all nodes in the original graph, as represented by the blue boxes at the bottom.

## D  FEATURE VISUALISATION

To visualise feature representation, we project node features from one class to 2 dimensions. It can be observed that the nodes tend to form multiple clusters in the feature space, exhibiting multi-modal feature distributions.

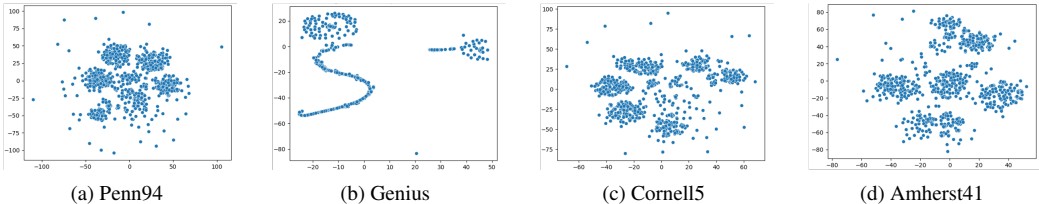

(a) Penn94      (b) Genius      (c) Cornell5      (d) Amherst41

Figure 5: Feature space visualization of several heterophilous datasets.

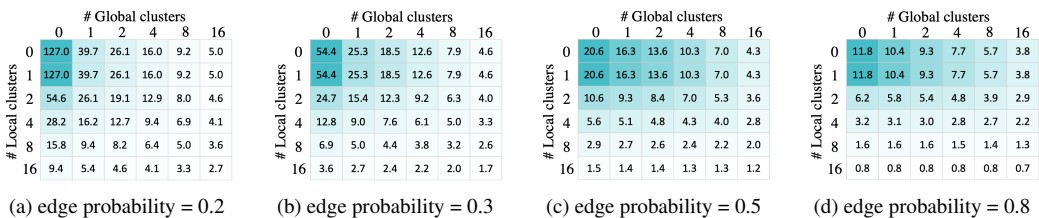

(a) edge probability = 0.2    (b) edge probability = 0.3    (c) edge probability = 0.5    (d) edge probability = 0.8

Figure 6: Total effective resistance heatmap of Erdos-Renyi random graphs at different sparsity levels. Number of nodes is 10 for all settings.

# E   EXPERIMENTS

## E.1   EFFECTIVE RESISTANCE ON RANDOM GRAPHS

Let $u$ and $v$ be vertices of $G$. The effective resistance between $u$ and $v$ is defined as

$$R_{u,v} = (1_u - 1_v)^T L^+ (1_u - 1_v),$$

where $1_v$ is the indicator vector of the vertex $v$ (Black et al., 2023). Let $A$ be the adjacency matrix and $D$ be the degree matrix. The Laplacian is $L = D - A$ and $L^+$ is the pseudoinverse of $L$. The total effective resistance ($R_{\text{tot}}$) of a graph is therefore the total sum of effective resistance between every pair of nodes.

We measure effective resistance ($R_{\text{tot}}$) in synthetic random graphs with different degrees of sparsity. Results in Fig. 6 show that both global and local cluster-nodes contribute to reducing effective resistance, as demonstrated by decreasing $R_{\text{tot}}$ values in both row and column directions. Additionally, the more drastic $R_{\text{tot}}$ decrease from the first to last column in heatmap (a) compared to heatmap (d) show that global cluster-nodes play a more pronounced role in reducing effective resistance at a higher edge sparsity setting.

## E.2   MORE ABLATION STUDIES

### E.2.1   AGGREGATION OPERATION IN SIMILARITY LOSS

Table 8: Effects of aggregator function in $\mathcal{L}_{\text{sim}}$.

| AGG | Penn94 | Amherst41 |
|------|--------------|--------------|
| MEAN | 86.13 (0.12) | 81.23 (1.34) |
| SUM | 85.84 (0.26) | 81.26 (1.58) |
| MAX | 86.69 (0.22) | 82.94 (1.59) |

After computing the similarity between a node and each of the multiple clusters from the same class, the choice of aggregation method is crucial. We evaluate the effectiveness of using the aggregation operator on Amherst41 and Penn94 datasets. Tab. 8 shows the effects of replacing the max aggregator with mean and sum in computing similarity loss. On both datasets, max outperforms both sum and mean, indicating the effectiveness of using max as the aggregation operation. Intuitively, taking the

average of all similarity scores (mean) is sub-optimal. mean tends to make the node embeddings closer to the average of all clusters belonging to a same class, undermining the purpose of using multiple clusters. Similar to mean, summing up all similarity scores (sum) is more powerful yet requires more data to learn. max selects the maximum similarity score to compute similarity loss and guides the node embeddings closer to one of the clusters, thus preserving the power of diversity in representation.

### E.2.2 ADDITIONAL ABLATION ON THE ORTHOGONALITY LOSS AND SIMILARITY LOSS

Table 9: Effects of Similarity and Orthogonality losses for datasets in Tab. 1.

| | Penn94 | Genius | Cornell5 | Amherst41 | US-election | Wisconsin | Cora | Citeseer | Pubmed |
|---|---|---|---|---|---|---|---|---|---|
| DC-GNN | 86.69 (0.22) | 91.70 (0.08) | 84.68 (0.24) | 82.94 (1.59) | 89.59 (1.60) | 91.67 (1.95) | 89.13 (1.18) | 77.93 (1.82) | 91.00 (1.28) |
| $(-)\mathcal{L}_{sim}, \mathcal{L}_{ortho}$ | 86.40 (0.25) | 91.68 (0.08) | 84.34 (0.22) | 81.75 (1.07) | 88.61 (1.57) | 89.06 (2.55) | 88.87 (1.23) | 77.27 (1.86) | 91.00 (1.61) |

Table 10: Effects of Similarity and Orthogonality losses on recent heterophilous datasets (Platonov et al., 2023).

| | Roman-empire | Amazon-ratings | Minesweeper | Tolokers | Questions |
|---|---|---|---|---|---|
| DC-GNN | 89.96 (0.35) | 51.11 (0.47) | 98.50 (0.21) | 85.88 (0.81) | 78.96 (0.60) |
| $(-)\mathcal{L}_{sim}, \mathcal{L}_{ortho}$ | 89.44 (0.72) | 50.32 (0.46) | 98.04 (0.19) | 84.94 (0.59) | 77.30 (0.98) |

We conduct ablation studies on all fourteen datasets we have used. As shown in Tab. 9 and Tab. 10, $\mathcal{L}_{ortho}$ and $\mathcal{L}_{sim}$ generally help to improve the scores across datasets by facilitating the clustering process. While $\mathcal{O}^{\lambda}_{cluster}$ is central to our model, the auxiliary losses $\mathcal{L}_{ortho}$ and $\mathcal{L}_{sim}$ play a more supportive role to facilitate the clustering process, likely by promoting distinct cluster representations and enhancing node-cluster alignment, as hypothesized. The modest performance uplift suggests that these losses make a positive, albeit limited, contribution, underscoring the dominant influence of $\mathcal{O}^{\lambda}_{cluster}$ in DC-GNN's superior performance.

### E.2.3 EFFECTS OF NODE FIDELITY TERM

Table 11: Dirichlet Energy (DE) with different $\beta$ values. Higher DE indicates increased node distinctiveness.

| | $\beta = 0.0$ | $\beta = 0.5$ | $\beta = 1.0$ |
|---|---|---|---|
| Wisconsin | 0.741288 | 0.963261 | 0.978465 |
| Citeseer | 0.110083 | 0.151699 | 0.206022 |
| Cora | 0.218658 | 0.294024 | 0.330234 |

The node fidelity term encourages the node embeddings to retain some information from the original node features, which serve as initial residual. This technique can also potentially help to alleviate oversmoothing as shown in Klicpera et al. (2018); Chen et al. (2020). To validate this, we conduct additional experiments to measure the normalized Dirichlet Energy (DE) (Karhadkar et al., 2022) for DC-GNN on Wisconsin, Cora and Citeseer, using the implementation from (Karhadkar et al., 2022).

We set $\beta$ to 0, 0.5 and 1 for each dataset to measure how increased weightage of the node fidelity term influences DE, while keeping $\alpha$ constant at 0.5. As observed in Tab. 11, DE positively correlates with $\beta$ on all datasets.

### E.3 RUNTIME EXPERIMENTS

We measure the average runtime of a `DC-MsgPassing` layer on the three largest datasets used in our experiments against Pytorch Geometric (Fey and Lenssen, 2019) implementation of GATConv and GCNConv. `DC-MsgPassing` takes less than 4x times GCN and is faster than GAT on these datasets. The results show that `DC-MsgPassing` is competitive in terms of runtime, confirming our complexity analysis.

Table 12: Average runtime and dataset statistics comparison on three large-scale datasets. Runtime results are in seconds.

|  | Penn94 | Cornell5 | Genius |
|---|---|---|---|
| # Nodes | 41,554 | 18,660 | 421,961 |
| # Edges | 1,362,229 | 790,777 | 984,979 |
| `DC-MsgPassing` | 0.00852 | 0.00820 | 0.00708 |
| GATConv | 0.01242 | 0.01685 | 0.01636 |
| GCNConv | 0.00220 | 0.00242 | 0.00207 |
| Multiples of GAT | 0.69x | 0.49x | 0.43x |
| Multiples of GCN | 3.88x | 3.38x | 3.42x |

## E.4 GRAPH-LEVEL TASKS

Table 13: Comparison between DC-GNN and baseline methods on Peptides-func and Peptides-struct datasets as reported in (Gutteridge et al., 2023).

| Model | PEPTIDES-FUNC
AP ↑ | PEPTIDES-STRUCT
MAE ↓ |
|---|---|---|
| GCN | 0.5930 (0.0023) | 0.3496 (0.0013) |
| GINE | 0.5498 (0.0079) | 0.3547 (0.0045) |
| GatedGCN | 0.5864 (0.0077) | 0.3420 (0.0013) |
| GatedGCN+PE | 0.6069 (0.0035) | 0.3357 (0.0006) |
| DIGL+MPNN | 0.6469 (0.0019) | 0.3173 (0.0007) |
| DIGL+MPNN+LapPE | 0.6830 (0.0026) | 0.2616 (0.0018) |
| MixHop-GCN | 0.6592 (0.0036) | 0.2921 (0.0023) |
| MixHop-GCN+LapPE | 0.6843 (0.0049) | 0.2614 (0.0023) |
| Transformer+LapPE | 0.6326 (0.0126) | 0.2529 (0.0016) |
| SAN+LapPE | 0.6384 (0.0121) | 0.2683 (0.0043) |
| GraphGPS+LapPE | 0.6535 (0.0041) | 0.2500 (0.0005) |
| NeuralWalker | 0.7096 (0.0078) | **0.2463** (0.0005) |
| DRew-GCN | 0.6996 (0.0076) | 0.2781 (0.0028) |
| DRew-GCN+LapPE | **0.7150** (0.0044) | 0.2536 (0.0015) |
| DRew-GIN | 0.6940 (0.0074) | 0.2799 (0.0016) |
| DRew-GIN+LapPE | 0.7126 (0.0045) | 0.2606 (0.0014) |
| DRew-GatedGCN | 0.6733 (0.0094) | 0.2699 (0.0018) |
| DRew-GatedGCN+LapPE | 0.6977 (0.0026) | 0.2539 (0.0007) |
| DC-GNN | 0.6850 (0.0075) | **0.2473** (0.0016) |

Table 14: Performance of DC-GNN against baselines on Mutag, Proteins and Enzymes. * indicates the best performing backbone(s) for the rewiring method.

| Rewiring | Model | Mutag | Proteins | Enzymes |
|---|---|---|---|---|
| None | GCN | 72.15 (2.44) | 70.98 (0.74) | 27.67 (1.16) |
| None | R-GCN | 69.25 (2.09) | 69.52 (0.73) | 28.60 (1.19) |
| None | GIN | 77.70 (3.60) | 70.80 (0.83) | 33.80 (1.12) |
| None | R-GIN | 83.05 (1.44) | 70.50 (0.81) | 39.12 (1.17) |
| None | PPGN (Maron et al., 2019) | 90.55 (8.7) | 77.20 (4.73) | - |
| None | CIN++ (Giusti et al., 2023) | 94.4 (3.7) | 80.5 (3.9) | - |
| PR-MPNN (Qian et al., 2023) | GIN* | **98.4** (2.4) | **80.7** (3.9) | - |
| DIGL (Gasteiger et al., 2019) | R-GIN* | 81.45 (1.49) | 71.31 (0.76) | 37.60 (1.20) |
| SDRF (Topping et al., 2021) | R-GIN* | 82.70 (1.78) | 70.70 (0.82) | 39.58 (1.33) |
| FoSR (Karhadkar et al., 2022) | R-GIN* | 86.15 (1.49) | 75.25 (0.86) | 45.55 (0.13) |
| GTR (Black et al., 2023) | R-GIN* | 86.10 (1.76) | 75.64 (0.74) | 50.03 (1.32) |
| DC-GNN | DC-GNN | 89.50 (3.11) | 77.95 (2.05) | **56.83** (4.20) |

Our work is motivated from the perspective of the node classification task. The clustering inductive bias aligns with node classification as the act of assigning a node to a cluster is in congruence with assigning a node to a class label. However, the alignment of the proposed clustering inductive bias with the graph-level tasks is somewhat unclear.

To investigate how DC-GNN performs on graph-level tasks, we conduct experiments on Peptides-func and Peptides-struct datasets from (Dwivedi et al., 2022) and Mutag, Proteins and Enzymes datasets from (Morris et al., 2020), with mixed results as shown in Tab. 13 and Tab. 14.

To better align with graph-level tasks, one possible direction for improvement is to share clusters across different graphs. For example, on molecule graphs, sharing clusters representing common sub-

structures such as functional groups across different molecule graphs could potentially be beneficial for molecule property prediction tasks.

### E.5 WHEN WILL GLOBAL CLUSTERING HELP?

we posit that global clustering is particularly beneficial when intra-class nodes exhibit strong clustering tendencies in feature space. One way of quantifying this is graph conductance. Graph conductance is a measure of how well-connected a subset of nodes is to the rest of the graph relative to its internal connectivity. Specifically, it measures the ratio of the number of edges that cross the boundary of a set to the minimum of the number of edges in the set or its complement. Formally, the formula for conductance $\Phi(S)$ of a subset $S$ of nodes is given by:

$$\Phi(S) = \frac{|\text{cut}(S, \overline{S})|}{\min(\text{vol}(S), \text{vol}(\overline{S}))},$$

where $|\text{cut}(S, \overline{S})|$ is the number of edges between the set $S$ and its complement $\overline{S}$ (the rest of the graph), $\text{vol}(S)$ is the sum of the degrees of the nodes in $S$, and $\text{vol}(\overline{S})$ is the sum of the degrees of the nodes in $\overline{S}$. Lower conductance values indicate that the set $S$ is well-clustered, meaning it has relatively few connections to the rest of the graph, suggesting a strong internal cohesion within the cluster.

To measure conductance in the feature space, we construct a k-nearest neighbor (k-NN) graph based on the node features. In this graph, each node is connected to its k nearest neighbors according to feature similarity, rather than graph topology. We then measure conductance on this k-NN graph, using the same formula as above. For our experiments, we set $k = 5$ and report the average conductance across classes when there are more than two classes.

Table 15: Comparison of original and k-NN graph conductance across datasets.

| Dataset | Type | Original Graph | k-NN Graph |
|---------|------|----------------|------------|
| US-election | Heterophilous | 0.6916 | 0.4796 |
| Penn94 | Heterophilous | 0.9557 | 0.3863 |
| Cornell5 | Heterophilous | 0.8864 | 0.3945 |
| Amherst41 | Heterophilous | 0.9058 | 0.4405 |
| Cora | Homophilous | 0.4016 | 0.6665 |
| Citeseer | Homophilous | 0.6221 | 0.9230 |
| Pubmed | Homophilous | 0.4478 | 0.2985 |

Tab. 15 compares the conductance of the original graph with that of the k-NN graph constructed purely from features for various datasets. Our findings indicate that heterophilous datasets typically exhibit much lower conductance on the k-NN graph. Conversely, homophilous datasets tend to show lower conductance on the original graph. However, it is crucial to note that this pattern is not universal. Not all heterophilous graphs may conform to this trend, nor do all homophilous graphs exhibit the opposite behavior. By leveraging graph conductance as an analytical tool, researchers and practitioners can make more informed decisions about the applicability of global clustering techniques to their specific graph datasets, potentially leading to improved performance in various graph learning tasks, such as node classification.

### E.6 DATASET DETAILS

We conduct experiments on fourteen datasets, a mix of small-scale and large-scale datasets. Eleven of them are non-homophilous, including: (1) Roman-empire, Amazon-ratings, Minesweeper, Tolokers, Questions (Platonov et al., 2023); (2) Penn94, Genius, Cornell5, Amherst41 (Lim et al., 2021); (3) Wisconsin (Pei et al., 2020); (4) a US election dataset (Jia and Benson, 2020). Three are homophilous citation networks: Cora, Citeseer and Pubmed (Pei et al., 2020). We use the original train/validation/test splits when they exist. Otherwise we follow the splits specified in (Platonov et al., 2023; Lim et al., 2021; Chen et al., 2020).

### E.6.1 DATASET DESCRIPTION

**Roman-empire, Amazon-ratings, Minesweeper, Tolokers and Questions** are five datasets proposed in Platonov et al. (2023) to better evaluate the performance of GNNs under heterophilous settings. The description of each dataset is as follows.

**Roman-empire** is based on the Roman Empire article from English Wikipedia. Each node in the graph represents one word in the text, and each edge between two words represents either one word following another word or if the two words are connected in the dependency tree. Node features is its FastText word embeddings. The task is to predict a node's syntactic role.

**Amazon-ratings** is based on the Amazon product co-purchasing network metadata. Nodes represent products and edges connect products frequently purchased together. Node features are the mean of FastText embeddings for words in product description. The task is to predict the class of products' ratings.

**Minesweeper** is a synthetic dataset inspired by the Minesweeper game. The graph is a regular 100x100 grid where each node is connected its eight neighboring nodes. 20% of the nodes are randomly assgined as mines. The node features are one-hot-encoded numbers of the neighboring mines. The task is to predict if the nodes are mines.

**Tolokers** is based on data from the Toloka crowdsourcing platform. Nodes represent workers who have participated in the selected projects, while edges connect two workers who work on the same task. Node features are based on worker's profile information and task performance. The task is to predict which workers have been banned.

**Questions** is based on question-answering data from website Yandex Q. Nodes represent users and edges connect an answer provider to a question provider. Node features are the mean of FastText word embeddings of user profile description, with an additional binary feature indicating users with no descriptions. The task is to predict if the users remain active on the website.

**Penn94, Cornell5 and Amherst41** (Lim et al., 2021) are friendship network datasets extracted from Facebook of students from selected universities from 2005. Each node in the datasets represent a student, while node label represents the reported gender of the student. Node features include major, second major/minor, dorm/house, year, and high school.

**Wisconsin** (Pei et al., 2020) is a web page dataset collected from the computer science department of Wisconsin Madison. In this dataset, nodes represent web pages and edges are hyperlinks between them. Feature vectors of nodes are bag-of-words representations. The task is to classify the web pages into one of the five categories including student, project, course, staff and faculty.

**Genius** (Lim et al., 2021) is a sub-network from website genius.com, a crowd-sourced website of song lyrics annotations. Nodes represent users while edges connect users that follow each other. Node features include expertise scores expertise scores, counts of contributions and roles held by users. Around 20% of the users are marked with a "gone" label, indicating that they are more likely to be spam users. The task is to predict which users are marked.

**US-election** (Jia and Benson, 2020) is a geographical dataset extracted from statistics of Unite States election of year 2012. Nodes represent US counties, while edges connect bordering counties. Node features include income, education, population etc. The task is a binary classification to predict election outcome.

**Cora, Citeseer and Pubmed** (Pei et al., 2020) are citation graphs, where each node represents a scientific paper and two papers are connected when a paper cites the other. Each node is labeled with the research field and the task is to predict which field the paper belongs to. All three datasets are homophilous.

### E.6.2 DATASET STATISTICS

Tab. 16 covers statistics of datasets in Tab. 1. Tab. 17 covers statistics of datasets in Tab. 2.

**Homophily matrix**   Homophily refers to the degree of similarity between connected neighboring nodes in terms of their features or labels. There are many types of homophily measures proposed, including edge homophily (Zhu et al., 2020b), node homophily (Pei et al., 2020), and improved edge

Table 16: Dataset statistics for Tab. 1.

| | Penn94 | Cornell5 | Amherst41 | Genius | US-election | Wisconsin | Cora | Citeseer | Pubmed |
|---|---|---|---|---|---|---|---|---|---|
| **Edge Hom.** | 0.47 | 0.47 | 0.46 | 0.61 | 0.83 | 0.21 | 0.81 | 0.74 | 0.80 |
| **Improved Edge Hom.** (Lim et al., 2021) | 0.046 | 0.09 | 0.05 | 0.08 | 0.54 | 0.094 | 0.766 | 0.627 | 0.664 |
| **# Nodes** | 41,554 | 18,660 | 2,235 | 421,961 | 3,234 | 251 | 2,708 | 3,327 | 19,717 |
| **# Edges** | 1,362,229 | 790,777 | 90,954 | 984,979 | 11,100 | 466 | 5,278 | 4,676 | 44,327 |
| **# Node Features** | 4814 | 4735 | 1193 | 12 | 6 | 1,703 | 1,433 | 3,703 | 500 |
| **# Classes** | 2 | 2 | 2 | 2 | 2 | 5 | 6 | 7 | 3 |

Table 17: Dataset statistics for Tab. 2.

| | Roman-Empire | Amazon-Ratings | Minesweeper | Tolokers | Questions |
|---|---|---|---|---|---|
| **Edge Hom.** | 0.05 | 0.38 | 0.68 | 0.59 | 0.84 |
| **Improved Edge Hom.** (Lim et al., 2021) | 0.01 | 0.12 | 0.009 | 0.17 | 0.08 |
| **# Nodes** | 22,662 | 24,492 | 10,000 | 11,758 | 48,921 |
| **# Edges** | 32,927 | 93,050 | 39,402 | 519,000 | 153,540 |
| **# Node Features** | 300 | 300 | 7 | 10 | 301 |
| **# Classes** | 18 | 5 | 2 | 2 | 2 |

homophily (Lim et al., 2021). Homophily matrix proposed in Lim et al. (2021) is an important metric, since it can better reflect class-wise homophily. The homophiliy matrix is defined as:

$$H_{c_1,c_2} = \frac{|(u,v) \in E : c_u = c_1, \ c_v = c_2|}{|(u,v) \in E : c_u = c_1|},$$

(22)

for classes $c_1$ and $c_2$, $H_{c_1 c_2}$ denotes the proportion of edges between from nodes of class $c_1$ to nodes of class $c_2$. A homophilous graph has high values on the diagonal entries of $H$.

Fig. 7 are the homophily matrices for three well-known homophilous datasets: Cora, Citeseer and Pubmed (Yang et al., 2016). High homophily is signified by the high numbers in diagonal cells, whereas values of non-diagonal cells are mostly less than 0.1. This is different from the homophily matrices of heterophilous datasets, where values of non-diagonal cells are similar or even higher than diagonal cells.

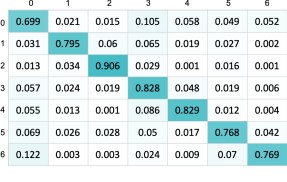
(a) Cora

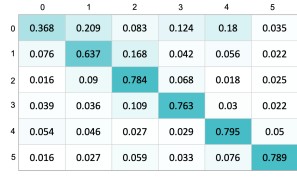
(b) Citeseer

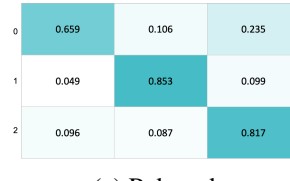
(c) Pubmed

Figure 7: Homophily matrix for three homophilous datasets.

We show in Fig. 8 the homophily matrices for some heterophilous datasets for comparison.

### E.7 MORE EXPERIMENT DETAILS

**Baselines** To comprehensively evaluate the effectiveness of our method, we compare it against various strong baselines following Platonov et al. (2023) and Li et al. (2022). This includes (1) graph-agnostic model MLP and ResNet (He et al., 2016), with two modified versions ResNet+SGC (Wu et al., 2019) and ResNet+adj (Zhu et al., 2021); (2) general-purpose GNN architectures: GCN(Kipf and Welling, 2016), Graph-SAGE (Hamilton et al., 2017), MixHop (Abu-El-Haija et al., 2019), GCNII (Chen et al., 2020), NaGphormer (Chen et al., 2022) and Exphormer (Shirzad et al., 2023); (3) GNN models that leverage attention-based aggregation: GAT (Veličković et al., 2017), Graph Transformer(GT) (Shi et al., 2020) and Neural Walker (Chen et al., 2024). Following Platonov et al. (2023), we also include two modified architectures GAT-sep and GT-sep, where ego- and neighbor-embeddings are aggregated separately. Following (Müller et al., 2023), we also include GPS$^{\text{GAT+Performer}}$ that achieves the best performance among GPS variants on most datasets as a baseline (Müller et al., 2023; Masters et al., 2022; Rampášek et al., 2022).(4) GNN models designed for heterophilous graphs: H$_2$GCN (Zhu et al., 2020b), CPGNN (Zhu et al., 2021), GPR-GNN (Chien et al., 2020), FSGNN (Maurya et al., 2022), FFAGCN (Bo et al., 2021), GBK-GNN (Du et al.,

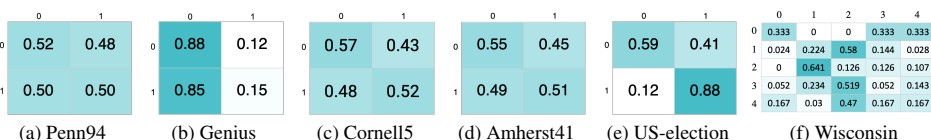

Figure 8: Homophily matrix for heterophilous datasets.

2022), Jacobi-Conv (Wang and Zhang, 2022), WRGAT (Suresh et al., 2021), GPR-GNN (Chien et al., 2020), GGCN (Yan et al., 2021), ACM-GCN (Luan et al., 2021), LINKX (Zhu et al., 2021), GloGNN/GloGNN++ (Li et al., 2022), GOAT (Kong et al., 2023), GMN (Behrouz and Hashemi, 2024), Diag-NSD (Bodnar et al., 2022), ACMP (Wang et al., 2022), CDE-GRAND (Zhao et al., 2023) and CDE-GraphBel (Zhao et al., 2023).

**Implementation details.** For global cluster-nodes, we use trainable lookup embeddings to initialize the embeddings. For local cluster-nodes, we fix the number of local clusters to be 2 for every ego-neighborhoods, and initialize the two cluster-node embeddings by the central node features and the average of neighboring node features respectively. In practice, for local clustering cost matrices M, we rescale and normalize the distance before running the Sinkhorn-Knopp algorithm for numerical stability. We apply non-linear activation function tanh to the messages.

**Training Settings** We conduct each experiment of DC-GNN using three distinct data splits and present the corresponding mean and standard deviation of the performance metrics. The experiments are executed on a single GPU. The GPUs are from various types—specifically, the V100, A100, GeForce RTX 2080, or 3090—based on their availability at the time the experiments are conducted. For optimization, we employ the Adam optimizer and undertake a grid search of hyperparameters, the specifics of which are in Tab. 18 and Tab. 19. Should the baseline results be publicly accessible, we directly incorporate them into our report. For the datasets where baseline results are missing (Cornell5, Amherst41, US-election), we reproduced them following the code in Li et al. (2022).

**Hyperparameters for DC-GNN** For experiments in Tab. 1 and Tab. 2, we fix some hyperparameters and perform grid search for other hyperparameters. To facilitate reproducibility, we document the details of the hyperparameters and search space in Tab. 18 and Tab. 19 respectively. $T^{\Omega}$ refer to the number of iterations of Sinkhorn–Knopp algorithm when solving $P^{\Omega}$, and $T^{\Gamma}$ is the number of Sinkhorn–Knopp iterations for solving $P^{\Gamma}$. We set $|\Omega|$ to be multiples of the number of classes in a dataset.

Table 18: Hyper-parameter search space of DC-GNN for datasets in Tab. 1.

| | Penn94 | Cornell5 | Amherst41 | Genius | US-election | Wisconsin | Cora | Citeseer | Pubmed |
|---|---|---|---|---|---|---|---|---|---|
| lr | 0.005 | 0.005 | 0.005 | 0.005 | 0.005 | 0.01 | 0.005 | 0.001, 0.002 | 0.005 |
| $\lambda$ | 2 | 2, 5 | 2 | 2 | 2,5 | 2 | 2 | 2 | 2 |
| $T^{\Omega}$ | 10,5,3 | 10,5,3 | 10,5,3 | 10,5,3 | 10,5,3 | 10,5,3 | 10,5,3 | 10,5,3 | 10,5,3 |
| $T^{\Gamma}$ | 5, 3, 1 | 5, 3, 1 | 5, 3, 1 | 5, 3, 1 | 5, 3, 1 | 5, 3, 1 | 5, 3, 1 | 5, 3, 1 | 5, 3, 1 |
| $|\Gamma_i|$ | 2 | 2 | 2 | 2 | 2 | 2 | 2 | 2 | 2 |
| $|\Omega|$ | 2,4, 8, 16, 30 | 2,4, 8, 16, 30 | 2,4, 8, 16, 30 | 2,4,8 | 2,4, 8, 16, 30 | 5, 10, 20 | 6,12,24,48 | 7, 14, 28 | 6, 12 |
| $\alpha$ | 0. 0.2, 0.5, 0.8, 1 | 0. 0.2, 0.5, 0.8, 1 | 0. 0.2, 0.5, 0.8, 1 | 0. 0.2, 0.5, 0.8 | 0.2, 0.5, 0.8 | 0.2, 0.5, 0.8 | 0.2, 0.5, 0.8 | 0.2, 0.5, 0.8 | 0.2, 0.5, 0.8 |
| $\beta$ | 0, 0.2, 0.5, 0.8 | 0, 0.2, 0.5, 0.8 | 0, 0.2, 0.5, 0.8 | 0, 0.2, 0.5, 0.8 | 0.2, 0.5, 0.8 | 0.2, 0.5, 0.8 | 0.2, 0.5, 0.8 | 0.2, 0.5, 0.8 | 0.2, 0.5, 0.8 |
| # layers in MLP | 1,2 | 1,2 | 1,2 | 1,2 | 1,2 | 1,2 | 1,2 | 1,2 | 1,2 |
| $L$ : # layers | 2, 5 | 2, 5 | 2, 5 | 2, 4,8,16 | 2, 5 | 2, 5 | 2,4,8,10 | 2,4,8 | 2,4,8 |
| $\omega_1$ | 0.001, 0.01 | 0.001, 0.01 | 0.001, 0.01 | 0.001,0.01 | 0.001, 0.01 | 0.001, 0.01 | 0, 0.001, 0.01, 0.05 | 0.001, 0.01 | 0.001, 0.01 |
| $\omega_2$ | 0.005, 0.05 | 0.005, 0.05 | 0.005, 0.05 | 0,0.005,0.05 | 0.005, 0.05 | 0.005, 0.05 | 0.005, 0.05, 0.08, 0.1 | 0.005, 0.05 | 0.005, 0.05 |
| epochs | 30 | 30 | 30 | 3000 | 500 | 200 | 200 | 50 | 500 |
| weight_decay | 5e-4 | 5e-4 | 5e-4 | 5e-4 | 5e-4 | 1e-3 | 5e-4 | 5e-4 | 5e-4 |
| aggregation | mean, sum | mean, sum | mean, sum | mean, sum | mean, sum | mean, sum | mean, sum | mean, sum | mean, sum |
| dropout | 0.5 | 0.5 | 0.5 | 0.5 | 0.5 | 0.5 | 0.5 | 0.5 | 0.5 |
| normalization | None | None | None | LN | None | None | None | None | None |
| hidden_channels | 16, 32, 64, 128 | 16, 32, 64, 128 | 16, 32, 64, 128 | 16, 32 | 16, 32, 64, 128 | 16, 32, 64, 128 | 16, 32, 64, 128 | 16, 32, 64, 128 | 16, 32, 64, 128 |

**Sensitivity test for hyperparameters** We conducted a sensitivity analysis on the hyperparameters $\alpha$, $\beta$, $\omega_1$ and $\omega_2$. As shown in Fig. 9, $\alpha$ is the most critical hyperparameter for our model. For both the US-election and Genius datasets, the optimal values fall between 0 and 1, indicating that balancing between local and global information is beneficial. More importantly, the sensitivity analysis of $\beta$, $\omega_1$ and $\omega_2$ reveals that model performance is stable in the neighbourhood of the optimal hyperparameters and not sensitive to small changes in the hyperparameter.

Table 19: Hyper-parameter search space of DC-GNN for datasets in Tab. 2.

| | Roman-Empire | Amazon-Ratings | Minesweeper | Tolokers | Questions |
|---|---|---|---|---|---|
| lr | 0.005 | 0.005 | 0.005 | 0.005 | 0.001 |
| $\lambda$ | 2 | 2 | 2 | 2 | 2 |
| $T^{\Omega}$ | 5 | 5 | 5 | 5 | 5 |
| $T^{\Gamma}$ | 3 | 3 | 3 | 3 | 3 |
| $|\Gamma_i|$ | 2 | 2 | 2 | 2 | 2 |
| $|\Omega|$ | 18 | 5, 10 | 2,4,8 | 8, 16 | 2,4,8 |
| $\alpha$ | 0.2, 0.5, 0.8 | 0.2, 0.5, 0.8 | 0.2, 0.5, 0.8 | 0.2, 0.5, 0.8 | 0.2, 0.5, 0.8 |
| $\beta$ | 0.2, 0.5, 0.8 | 0.2, 0.5, 0.8 | 0.2, 0.5, 0.8 | 0.2, 0.5, 0.8 | 0.2, 0.5, 0.8 |
| # layers in MLP | 1,2 | 1,2 | 1,2 | 1,2 | 1,2 |
| $L$ : # layers | 2, 4, 8,16,20 | 2, 4, 8 | 2, 4, 8 | 2, 4, 8 | 2, 4, 8, 16 |
| $\omega_1$ | 0.001, 0.01 | 0.001, 0.01 | 0.001, 0.01 | 0.02 | 0.001, 0.01 |
| $\omega_2$ | 0.005, 0.05 | 0.005, 0.05 | 0.005, 0.05 | 0.001 | 0.001 |
| epochs | 3000 | 3000 | 3000 | 3000 | 3000 |
| weight_decay | 5e-4 | 5e-4 | 5e-4 | 5e-4 | 5e-4 |
| aggregation | mean, sum | mean, sum | mean, sum | mean, sum | mean, sum |
| dropout | 0.2 | 0.2 | 0.2 | 0.2 | 0.2 |
| normalization | None, LN | None, LN | None, LN | None, LN | None, LN |
| hidden_channels | 16, 32, 64 | 16, 32, 64 | 16, 32, 64 | 16, 32, 64 | 16, 32, 64 |

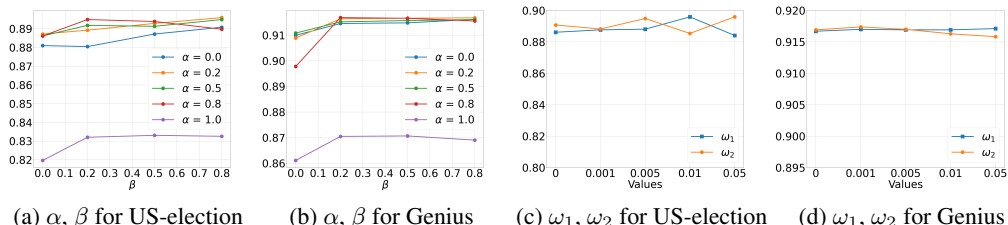

(a) $\alpha$, $\beta$ for US-election    (b) $\alpha$, $\beta$ for Genius    (c) $\omega_1$, $\omega_2$ for US-election    (d) $\omega_1$, $\omega_2$ for Genius

Figure 9: Sensitivity test of $\alpha$, $\beta$, $\omega_1$, $\omega_2$. x-axis is the varying value of hyperparameters, while y-axis is model performance.

We also conducted sensitivity analysis on the hyperparameters $|\Omega|$ and $\lambda$. As shown in Fig. 10, model performance is stable and insensitive to changes in both $|\Omega|$ and $\lambda$. However, we observe that large $\lambda$ can lead to numerical stability issues.

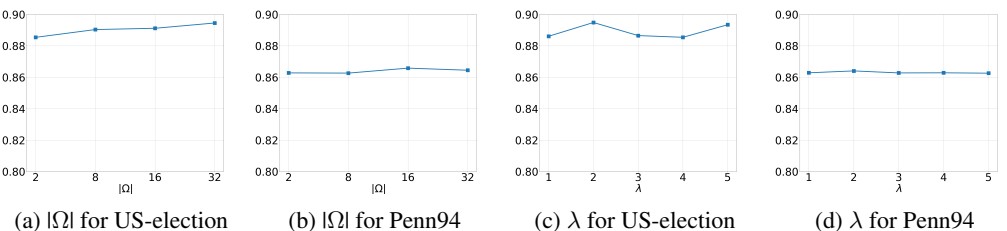

(a) $|\Omega|$ for US-election    (b) $|\Omega|$ for Penn94    (c) $\lambda$ for US-election    (d) $\lambda$ for Penn94

Figure 10: Sensitivity test of $|\Omega|$ and $\lambda$. x-axis is the varying value of hyperparameters, while y-axis is model performance.

**Training time and loss curve** We have measured the average training time per epoch of our method on dataset Penn94, Cornell5 and Genius in Tab.20, Tab.21 and Tab.22 respectively. The training time ranges from 3x to 5x compared to GCN.

Training loss curves on dataset Penn94 and US-election can be found in Fig. 11.

**Hyperparameters for baselines** We used the code provided by GloGNN (Li et al., 2022) to reproduce the baseline results for dataset Cornell5, Amherst41, and US-election. The grid search space for hyper-parameters are listed below. Note that some hyper-parameters only apply to a subset of baselines. All other baselines results are obtained from Li et al. (2022) and Platonov et al. (2023).

- MLP: hidden dimension $\in \{16, 32, 64\}$, number of layers $\in \{2, 3\}$. Activation function is ReLU.
- GCN: lr $\in \{.01, .001\}$, hidden dimension $\in \{4, 8, 16, 32, 64\}$. Activation function is ReLU.

Table 20: Training times of our method against GCN on Penn94.

|  | Training Time | Multiples of GCN |
|---|---|---|
| GCN | 0.0637 | - |
| $|\Omega| = 2$ | 0.2880 | 4.51x |
| $|\Omega| = 4$ | 0.2888 | 4.53x |
| $|\Omega| = 8$ | 0.2933 | 4.60x |
| $|\Omega| = 16$ | 0.3007 | 4.71x |

Table 21: Training times of our method against GCN on Cornell5.

|  | Training Time | Multiples of GCN |
|---|---|---|
| GCN | 0.0602 | - |
| $|\Omega| = 2$ | 0.2201 | 3.65x |
| $|\Omega| = 4$ | 0.2056 | 3.41x |
| $|\Omega| = 8$ | 0.2161 | 3.58x |
| $|\Omega| = 16$ | 0.2332 | 3.87x |

Table 22: Training times of our method against GCN on Genius.

|  | Training Time | Multiples of GCN |
|---|---|---|
| GCN | 0.1892 | - |
| $|\Omega| = 2$ | 0.8467 | 4.47x |
| $|\Omega| = 4$ | 0.8911 | 4.70x |
| $|\Omega| = 8$ | 0.9211 | 4.86x |
| $|\Omega| = 16$ | 1.0756 | 5.68x |

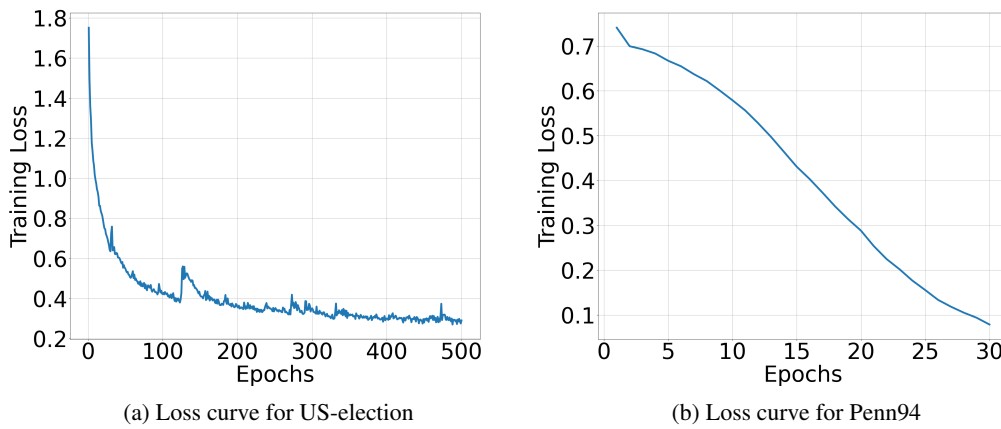

(a) Loss curve for US-election  (b) Loss curve for Penn94

Figure 11: Training loss curves for Penn94 and US-election.

- GAT: lr $\in \{.01, .001\}$. hidden channels $\in \{4, 8, 12, 32\}$ and gat heads $\in \{2, 4, 8\}$. number of layers $\in \{2\}$. We use the ELU as activation.

- MixHop: hidden dimension $\in \{8, 16, 32\}$, number of layers $\in \{2\}$.

- GCNII: number of layers $\in \{2, 8, 16, 32, 64\}$, strength of initial residual connection $\alpha \in \{0.1, 0.2, 0.5\}$, hyperparameter for strength of the identity mapping $\theta \in \{0.5, 1.0, 1.5\}$.

- H$_2$GCN: hidden dimension $\in \{16, 32\}$, dropout $\in \{0, .5\}$, number of layers $\in \{1, 2\}$. Model architecture follows Section 3.2 of Zhu et al. (2020b).

- WRGAT: lr $\in \{.01\}$, hidden dimension $\in \{32\}$.

- GPR-GNN: lr $\in \{.01, .05, .002\}$, hidden dimension $\in \{16, 32\}$.

- GGCN: lr $\in \{.01\}$, hidden channels $\in \{16, 32, 64\}$, number of layers $\in \{1, 2, 3\}$, weight decay $\in \{1e^{-7}, 1e^{-2}\}$, decay rate $\in \{0, 1.5\}$, dropout rate $\in \{0, .7\}$,

- ACM-GCN: lr $\in \{.01\}$, weight decay $\in \{5e^{-5}, 5e^{-4}, 5e^{-3}\}$, dropout $\in \{0.1, 0.3, 0.5, 0.7, 0.9\}$, hidden channels $\in \{64\}$, number of layers $\in \{2\}$, display step $\in \{1\}$.

- LINKX: hidden dimension $\in \{16, 32, 64\}$, number of layers $\in \{1, 2\}$. Rest of the hyperparameter settings follow Lim et al. (2021).

- GloGNN++: lr $\in \{.001, .005, .01\}$ , weight decay $\in \{0, .01, .1\}$, dropout $\in \{0, .5, .8\}$, hidden channels $\in \{128, 256\}$, number of layers $\in \{1, 2\}$, $\alpha \in \{0, 1\}$, $\beta \in \{0.1, 1\}$, $\gamma \in \{0.2, 0.5, 0.9\}$, $\delta \in \{0.2, 0.5\}$, number of normalization layers $\in \{1, 2\}$, orders $\in \{1, 2, 3\}$.

