# OpenReview forum: "Differentiable Cluster Graph Neural Network"
_ICLR.cc/2025/Conference — Submitted to ICLR 2025_

### Official Review · Reviewer_wyzZ · 2024-10-27

**Soundness:** 3
**Presentation:** 2
**Contribution:** 2
**Rating:** 5
**Confidence:** 4

**Summary:**

This paper incorporates a clustering inductive bias, combining nodes with "cluster nodes" in a bipartite structure and leveraging optimal transport for differentiable clustering. The framework iteratively optimizes node and cluster-node embeddings and integrates clustering directly within message passing, effectively capturing both global and local patterns.

**Strengths:**

1. This work integrates clustering into message passing, allowing for both global and local pattern capture, which is effective for diverse graphs.
2.  This work demonstrates effectiveness on multiple datasets.

**Weaknesses:**

1. The author claims that facilitating long-range interactions across distant nodes. Thus it should be tested on long-range graph datasets in [1].
2. The proposed method does not achieve the best performance in all cases in Table 2. More analysis on why performing poorly should be added.
3. [2] and [3] are related to this work. I think quoting them is needed.
4. The OT-based clustering and iterative optimization may add implementation complexity.

[1] Dwivedi, Vijay Prakash, et al. "Long range graph benchmark." Advances in Neural Information Processing Systems 35 (2022): 22326-22340.
[2] Kosmala, Arthur, et al. "Ewald-based long-range message passing for molecular graphs." International Conference on Machine Learning. PMLR, 2023.
[3] Chen, Dexiong, Till Hendrik Schulz, and Karsten Borgwardt. "Learning Long Range Dependencies on Graphs via Random Walks." arXiv preprint arXiv:2406.03386 (2024).

**Questions:**

See Weakness.

---

> ### Author Response · Authors · 2024-11-22
>
> Thank you very much for your valuable feedback. We highly appreciate the comments and suggestions for improvement and will incorporate them into our revised manuscript. The reply to the questions is enclosed below.
>
> >The author claims that facilitating long-range interactions across distant nodes. Thus it should be tested on long-range graph datasets in [1].
>
> **Ans**: In Table 13 of the Appendix, we have evaluated our methods on Peptides-func and Peptides-struct datasets from long range graph benchmarks(LRGB) [1]. More discussion can be found in Appendix D.4. Note that our current formulation of the bipartitie graph and implementation of message passing does not incorporate edge features which exist in LRGB datasets, and so we have removed the edge feature information in our LRGB experiments.
>
>
> >The proposed method does not achieve the best performance in all cases in Table 2. More analysis on why performing poorly should be added.
>
> **Ans**: We consider our method’s performance to be competitive, as it achieves the best results on 12 out of 14 datasets and ranks third-best on another. Our method remains competitive even when compared to the most recent baselines.
>
> The variation in performance across different datasets may stem from the degree of alignment between the global and local clustering inductive biases and the specific task. This alignment is challenging to quantify due to the latent nature of feature representations. We acknowledge this limitation and identify it as an important future research direction.
>
>
> >[2] and [3] are related to this work. I think quoting them is needed.
>
> **Ans**: Thank you for your suggestion. We will ensure that [2] and [3] are properly cited. Additionally, we observed that [3] conducted experiments on some of the same datasets as our study. To provide a more comprehensive comparison, we have incorporated [3] as an additional baseline in Table 2 and Table 13.
>
>
> >The OT-based clustering and iterative optimization may add implementation complexity.
>
> **Ans**: One of the key contributions of our work is introducing a simple yet effective solution to optimize OT-based clustering by **leveraging straightforward matrix scaling and iterative message passing** on graphs.
>
> While OT-based clustering is often considered challenging to implement, our approach simplifies it by updating the assignment matrices using basic matrix scaling via the Sinkhorn–Knopp algorithm. This is made feasible by adopting the entropy-regularized version of the objective function, which significantly reduces implementation complexity.
>
> Additionally, we seamlessly translate the iterative optimization steps into classical iterative message-passing steps. This design choice ensures that no additional complexity is introduced, making our method both practical and easy to implement.

---

> > ### Comment · Reviewer_wyzZ · 2024-11-25
> >
> > Why does the performance of [3] outperform the proposed method once again?

---

> ### Author Response · Authors · 2024-11-26
>
> Thanks for your question and engagement.
>
> In Table 2, we observe that DC-GNN is outperformed by Neural Walker[3] on 2 out of 5 datasets , while DCGNN maintains competitive advantage on the other 3. **This variation in performance could be arguably attributed to the different inductive biases inherent in the two methods.**
>
>
> According to the No-Free-Lunch Theorem [1], no single algorithm performs best on all possible problems/datasets. The effectiveness of an algorithm depends on how well its inductive bias aligns with the underlying structure of the specific problem.
>
> - **Neural Walker's Inductive Bias**: Neural Walker leverages random walks to capture sequence information within graphs. By treating random walks as sequences and employing positional encodings on nodes within these sequences, it learns sequence embeddings that capture multi-hop interactions and long-range dependencies. This approach can be more expressive than the 1-WL test and is particularly effective on datasets where such multi-hop information is crucial.
>
> - **DC-GNN's Inductive Bias**: DC-GNN is designed to capture clustering patterns in the latent space. It focuses on connecting similar nodes while disconnecting dissimilar ones, effectively capturing cluster structures within the data, both globally for long range information and locally for heterophilous neighborhood. This makes DC-GNN especially suited for datasets where such cluster patterns are prominent and more informative than the explicit graph connectivity.
>
>
> **Orthogonality**: We also note that the inductive biases of DC-GNN and NeuralWalker are orthogonal. It's possible to consider integrating the two inductive biases. For instance, DC-GNN could be used as the MPNN component in the main update function(Eqn.(3)) of Neural Walker. This combination could leverage the strengths of both methods, although exploring this integration is beyond the scope of our current work.
>
> We appreciate the opportunity to discuss these nuances and believe that both methods contribute valuable perspectives to graph representation learning.
>
> **References:**
>
> [1] No-Free-Lunch Theorem: Wolpert, D. H., & Macready, W. G. (1997). No free lunch theorems for optimization. IEEE Transactions on Evolutionary Computation, 1(1), 67-82.

---

> > ### Author Response · Authors · 2024-11-29
> >
> > Dear Reviewer wyzZ,
> >
> > We sincerely appreciate your valuable comments and hope that our responses have effectively addressed your concerns. If you have any additional questions or issues, we'd be happy to discuss them with you further. As the discussion period is coming to a close, we look forward to hearing from you again.

---

### Official Review · Reviewer_29US · 2024-10-29

**Soundness:** 2
**Presentation:** 1
**Contribution:** 2
**Rating:** 8
**Confidence:** 3

**Summary:**

The paper proposes a differentiable cluster GNN to deal with the long-rage information propagation and local heterophilous neighborhood aggregation problem in graph domain. Specifically, the framework considers a clustering inductive bias into the message propagation by using additional cluster-nodes. Besides, authors adopt an iterative process to optimize the cluster assignments and node/cluster-node embeddings. Extensive experimental results show the effectiveness of the proposed framework.

**Strengths:**

1.	The paper addresses two critical challenges—over-squashing and heterophilous neighborhood aggregation with a unified framework.
2.	The iterative optimization with soft cluster assignment makes it possible to learn both node and cluster-node embeddings efficiently.
3.	Experimental results are sufficient to demonstrate the effectiveness of the proposed model.

**Weaknesses:**

1.	The complexity analysis is very rough, the preprocessing of constructing bipartite graph may cost time, so it would be better to supplement the complexity of preprocessing data.
2.	The introduction of so-called “cluster nodes” is basically assigning a pseudo label to each node, which is quite common among methods on graph heterophily.
3.	The novelty of directly integrate clustering into the message-passing mechanism is somehow weak.

**Questions:**

1.	The authors mention “cluster patterns” many times, but it is unclear what the definition of cluster patterns is?
2.	How does the construction of the bipartite graph scale for extremely large datasets?
3.	The paper uses soft cluster assignments but it lacks the sensitivity analysis of the entropic regularization \lambda and the number of clusters?
4.	From the ablation studies, the contributions of local and global clustering vary. It would be better to provide further intuition or guidelines on when one should prioritize local or global clusters for new datasets.
5.	As the bipartite graph grows with the addition of cluster-nodes, what are the memory implications?

---

> ### Author Response · Authors · 2024-11-22
>
> Thank you very much for your valuable feedback. We highly appreciate the comments and suggestions for improvement and will incorporate them into our revised manuscript. The reply to the questions is enclosed below.
>
> Weaknesses:
> >The complexity analysis is very rough, the preprocessing of constructing bipartite graph may cost time, so it would be better to supplement the complexity of preprocessing data.
>
> **Ans**: For the preprocessing of bipartite graph construction, the time needed is negligible, since we only need to modify the adjacency matrix (edge_index in Pytorch Geometric), and initialize the embeddings for cluster-nodes. It is linear in number of nodes and doesn't add to the training cost at each step.
>
> >The introduction of so-called “cluster nodes” is basically assigning a pseudo label to each node, which is quite common among methods on graph heterophily.
>
> **Ans**: We understand the reviewer may be refering to approaches like [1]. If the reviewer meant other types of approaches, we would appreciate clarification.
>
> Our approach is fundamentally different from assigning a pseudo-label to each node (as in [1]) in several key ways:
>
> 1. First, we assign multiple cluster-nodes per class based on the insight that datasets tend to exhibit multi-modal feature distribution even within one class (we added Figure 5 for visualization). A single cluster per class would be inaccurate or insufficient, as it may only represent an average or a dominant cluster.
>
> 1. Second, corresponding to our finding that cluster patterns can arise at both global and local levels, we are the first to devise both global and local cluster-nodes that connect to global neighborhood and local ego-neighborhood respectively.
>
> 1. Third, cluster-node are not just nodes with assigned labels, importantly the embeddings of cluster-nodes are learned in a **principled way to have physical meaning that represents cluster centroids**, which significantly differs from other approaches and, to our knowledge, has not been explored to tackle heterophily.
>
> Reference:
> [1] Dai, Enyan, et al. "Label-wise graph convolutional network for heterophilic graphs." Learning on Graphs Conference. PMLR, 2022.
>
> >The novelty of directly integrate clustering into the message-passing mechanism is somehow weak.
>
> **Ans**：We respectfully disagree. Direct integration of clustering into the message passing introduces several meaningful advancements:
>
> 1. **End-to-end design**: Previous methods in graph learning that incorporated clustering treated it as a separate component, limiting their ability to leverage the benefits of end-to-end learning. Our direct integration of clustering into the message-passing mechanism addresses the issue of non-differentiability of clustering and enables seamless end-to-end optimization.
> 1. **Principled framework**: Our approach is developed in principled manner with additional convergence properties.
> 1. **Unified inductive bias in training and inference**: Importantly, due to this direct integration into the message passing mechanism, the clustering inductive-bias is applied both in learning and inference phase. This is unlike the models which introduce the clustering bias with an auxillary loss function, which cannot enforce it during test time.

---

> > ### Author Response · Authors · 2024-11-22
> >
> > > The authors mention “cluster patterns” many times, but it is unclear what the definition of cluster patterns is?
> >
> > **Ans**: We use the term 'cluster patterns' to describe groupings in the data where nodes exhibit shared characteristics. To provide greater clarity, we have included additional visualization figures in the revised version (Figure 5 in Appendix D) illustrating the feature space of the nodes. These figures reveal that nodes often form multiple distinct clusters, a phenomenon we refer to as 'cluster patterns.'
> >
> > > How does the construction of the bipartite graph scale for extremely large datasets?
> >
> > **Ans**: We have run our experiments on large scale datasets with millions of edges and 0.4 million nodes (Penn94, Genius, we report dataset statistics in App.D.5.2).
> >
> > Extremely large dataset that takes hundreds of GB memory for dataset alone is beyond the scope of this work. Scaling to these datasets would require techniqes such as neighborhood sampling, mini-batching, parallel processing. Nonetheless, we are optimistic for the scalability of our methods for the following reasons:
> > - The time required for bipartitie graph construction is insignificant (only involves modifying the adjacency matrix and initializing the embeddings for cluster-nodes), and doesn't add to the training cost at each step
> > - The memory consumption is linear with respect to a standard GNN (see the answer to the last question).
> > - Training time is only 3 to 5 times more than a classical GCN (added experiments in Table 20, 21, 22).
> >
> > > The paper uses soft cluster assignments but it lacks the sensitivity analysis of the entropic regularization \lambda and the number of clusters?
> >
> > **Ans**: Following your suggestion, we conducted additional sensitive analysis on the number of clusters (|$\Omega$|) and the entropic regularization ($\lambda$). Figure 10 in App.E.7 show that the model performance is stable in the neighbourhood of the optimal hyperparameters and not sensitive to small changes.
> >
> > > From the ablation studies, the contributions of local and global clustering vary. It would be better to provide further intuition or guidelines on when one should prioritize local or global clusters for new datasets.
> >
> > **Ans**: Thanks for your suggestion. This is a valuble insight that stimulates further reflection on our part.
> > For local clusters, datasets with heterophilous neighborhood would benefit.
> > For global clusters, we posit that they are particularly beneficial when intra-class nodes exhibit strong clustering tendencies in feature space. One way of quantifying this is graph conductance, which measures how well-connected a subset of nodes is to the rest of the graph relative to its internal connectivity. Specifically, it measures the ratio of the number of edges that cross the boundary of a set to the minimum of the number of edges in the set or its complement. Lower conductance values indicate that the subset of the graph is well-clustered, as it has relatively few connections to the rest of the graph, suggesting strong internal cohesion within the cluster.
> >
> > To measure conductance in the feature space, we construct a k-nearest neighbor (k-NN) graph based on the node features. In this graph, each node is connected to its k nearest neighbors according to feature similarity, rather than graph topology. We then measure conductance on this k-NN graph and the original graph. The results indicate that Penn94, Cornell5 and Amherst41 have most reduction of graph conductance on k-NN graph and would probably benefit from global clustering.
> >
> > | **Dataset**    | **Type**        | **Original Graph** | **k-NN Graph** |
> > |-----------------|-----------------|--------------------|----------------|
> > | US-election     | Heterophilous   | 0.6916             | 0.4796         |
> > | Penn94          | Heterophilous   | 0.9557             | 0.3863         |
> > | Cornell5        | Heterophilous   | 0.8864             | 0.3945         |
> > | Amherst41       | Heterophilous   | 0.9058             | 0.4405         |
> > | Cora            | Homophilous     | 0.4016             | 0.6665         |
> > | Citeseer        | Homophilous     | 0.6221             | 0.9230         |
> > | Pubmed          | Homophilous     | 0.4478             | 0.2985         |
> >
> > More detailed discussion is added in App.E.5.
> >
> > > As the bipartite graph grows with the addition of cluster-nodes, what are the memory implications?
> >
> > **Ans**: The memory complexity is $O(|\mathcal{V}| + |\mathcal{C}|)$ = $O(|\mathcal{V}| + |\Omega| + \sum_{i \in \mathcal{V}} |\Gamma_i|)$ = $O(|\Gamma_i||\mathcal{V}|)$. Here, $|\mathcal{V}|$ is the number of nodes, $|\Omega|$ (the number of global cluster-nodes) is significantly smaller than $|\mathcal{V}|$, and $|\Gamma_i|$ (the number of local cluster-nodes per node) is a small constant in practice. Therefore, it is linear w.r.t the number of nodes, linear to a classical GNN. We have added this in the complexity analysis of the revised version.

---

> > > ### Comment · Reviewer_29US · 2024-11-25
> > >
> > > Thanks for the detailed response. I've read all your reply and the newest version of the paper together with the appendix. The experiments of sensitivity analysis of the entropic regularization seem to indicate that the model is not sensitive to the parameter settings. Maybe you can discuss it more when you have time. For now, all my concerns are addressed. I will increase my rating.

---

> > > > ### Author Response · Authors · 2024-11-25
> > > >
> > > > Thank you very much! We greatly appreciate your insightful comments and the time you spent reviewing our work. Your recognition of the merits of our research means a lot to us, and your feedback has been invaluable in improving our manuscript.

---

### Official Review · Reviewer_dTqj · 2024-11-03

**Soundness:** 2
**Presentation:** 2
**Contribution:** 2
**Rating:** 3
**Confidence:** 3

**Summary:**

This paper proposes a GNN that uses cluster-based messaging passing to address the over-squashing and heterophily problems in graph representation learning.The author claims that their cluster-based method (DC-GNN) can handle both long-range dependencies and heterophilous aggregation, by projecting graphs and node neighbors into a bipartite graph of global clusters and local clusters. DC-GNN is different than the current methods by taking the bipartite graph as input instead of using the adjacency or graph Laplacian in the network.

**Strengths:**

1. Converting the graph structure (adjacency or graph Laplacian) into a bipartite graph as GNN input is interesting.

**Weaknesses:**

1. No model access, no code access. (major)

2. Most baseline results reported in Table1 do not align with the results reported in original papers. Where did you get the baseline results? If you run the baselines, please provide codes/loggers or any proof. If not, please cite the sources that you used. (major)

3. Overall contribution is not much, not so exciting. (minor)

**Questions:**

see weaknesses

---

> ### Author Response · Authors · 2024-11-23
>
> We appreciate your effort in reviewing our paper. Please find our responses below, which we hope adequately address your concerns.
>
> > No model access, no code access. (major)
>
> **Ans**: As the code is proprietary, we require internal clearance before making it publicly accessible. To support the review process, we have shared a private link to the code for your review. We hope this addresses your concerns and are happy to assist further if needed. Please let us know if there is anything else we can provide to facilitate the process.
>
> We have also added a reproducibility statement before the references to provide pointers to sections of the paper on reproducibility.
>
> > Most baseline results reported in Table1 do not align with the results reported in original papers. Where did you get the baseline results? If you run the baselines, please provide codes/loggers or any proof. If not, please cite the sources that you used. (major)
>
> **Ans**: As stated in Appendix (line 1505 to 1508), all baseline results in Table 1 are obtained from Li et al. (2022), except for dataset Cornell5, Amherst41 and US-election where the original paper does not contain the baseline results. We have double checked and verified that baseline results are identical to those reported in Li et al. (2022)
>
> For dataset Cornell5, Amherst41 and US-election, we reproduced the results using the official repo (https://github.com/RecklessRonan/GloGNN) of Li et al. (2022) due to absence of such results in their paper, as stated in Appendix (line 1505 to 1508). To faciliate reproducibility, we have also provided the hyperparameter search space for these datasets, as stated in Appendix (line 1509 to 1511, 1566 to 1587).
>
> Reference:
>
> Xiang Li, Renyu Zhu, Yao Cheng, Caihua Shan, Siqiang Luo, Dongsheng Li, and Weining Qian. Finding global homophily in graph neural networks when meeting heterophily. ICML 2022.
>
> > Overall contribution is not much, not so exciting. (minor)
>
> **Ans**: While we value the reviewers opinion, we respectfully disagree. **Our work provides a novel way of addressing the issues of heterophily and long-range interactions in a principled way, integerating the clustering appraoch within message passing.** We believe this provides value to the understanding of these issues and provides a new perspective to viewing the ways to solve them, which future works can build upon.
> To list down the main contributions below.
> 1. **Unified clustering-based approach** tackles two issues in graph learning, - (1) to capture the long range interactions while mitigating oversquashing, and (2) to effectively aggregate local information in heterophilic neighbourhoods, where connected nodes in the graph are likely to be dissimilar.
> 2. We leverage Optimal Transport theory in the formulation of clustering based objective function, that involves both global and local clustering terms.  We derive **novel closed-form message passing functions** between nodes and cluster-centroids, **as optimization solutions for the entropic-regularized clustering based objective function**. Our DC-GNN is then an optimization framework which is **end-to-end differentiable**.
> 3. We prove the convergence of our algorithm and show strong empirical results across a variety of benchmark datasets.

---

> > ### Author Response · Authors · 2024-11-29
> >
> > Dear Reviewer dTqj,
> >
> > We hope we have addressed all of your concerns and are eager to engage in further discussion to resolve any remaining issues you may have. As the discussion period is coming to a close, we look forward to hearing from you soon. Thank you for your time and consideration.

---

### Official Review · Reviewer_6te7 · 2024-11-04

**Soundness:** 2
**Presentation:** 4
**Contribution:** 2
**Rating:** 6
**Confidence:** 3

**Summary:**

This paper presents a differentiable end-to-end clustering-based graph neural network for node classification tasks. The proposed model attempts to address over-squashing and heterogeneity. The model is carefully designed to make it differentiable and the authors provide theoretical guarantees on convergence and time complexity.

**Strengths:**

1. The paper is clearly written and easy to read.
2. The proposed end-to-end differentiable model is convincing and the proposed model is trying to solve the important problems encountered in graph representation learning models such as over-squashing and heterophily.
3. Experimental results show that the model works well and experimental details are provided in the appendix.

**Weaknesses:**

1. The model has multiple hyperparameters, which is very confusing for the potential user of the proposed model to select the optimal hyperparameters.
2. The proposed model seems difficult to train and converge despite Theorem 3.3 can provide some guarantee for convergence. I'm not sure if the model can converge only in a very narrow range of hyperparameters, and the code is not open-sourced.
3. $|\Omega|$ can not be removed from asymptotic time complexity $O(T|\mathcal{V}||\Omega|)$ simply because of $|\Omega| \ll |\mathcal{V}|$. ). I suggest that the authors keep the original complexity with these hyperparameters, and then provide a simplified complexity when these hyperparameters are considered as constants. I suggest that the authors report in their experiments the training time and running time of DCGNN in comparison with classical models such as GCN.
4. For this reason and concerns about convergence speed, I am worried about the actual training-to-convergence time might be longer than expect. I note that run times are provided in Appendix D.3, but I would expect the authors to report training times and loss curves.

**Questions:**

1.  How to choose model hyper-parameters in practice?
2. I expect the authors to provide the training time and loss curves for the model.

---

> ### Author Response · Authors · 2024-11-22
>
> Thank you very much for your valuable feedback. We highly appreciate the comments and suggestions for improvement and will incorporate them into our revised manuscript. The reply to the questions is enclosed below.
>
>
> > How to choose model hyper-parameters in practice?
>
> **Ans**: While the model has multiple hyperparameters, the most critical ones are $\alpha$ and $\beta$, which govern the clustering-based objective function $O_{cluster}$. Other key hyperparameters include the number of iterations (corresponding to the number of layers in the GNN) and the hidden dimension, similar to other GNNs.
>
> Experiments in Section 4.5.2, along with the new sensitivity analysis for other hyperparameters in Appendix E.7 (Figure 10) show that the model performance is stable in the neighbourhood of the optimal hyperparameters and not sensitive to small changes.
>
>
> > The proposed model seems difficult to train and converge despite Theorem 3.3 can provide some guarantee for convergence. I'm not sure if the model can converge only in a very narrow range of hyperparameters,
>
> **Ans**: After reviewing the experimental logs, we found that the model successfully converges for all hyperparameter configurations explored in 12 out of the 14 datasets. For the Cornell5 dataset, the model converges for 82% of the hyperparameter configurations we tested. For the Minesweepers dataset, the model converges in all cases except when $\alpha = 1$.
>
> >   the code is not open-sourced.
>
> **Ans**: The code is proprietary, and we need internal clearance to release it publicly.
>
> > Complexity Analysis: I suggest that the authors keep the original complexity with these hyperparameters, and then provide a simplified complexity when these hyperparameters are considered as constants.
>
> **Ans**: Thanks for your suggestion. We have revised it in the paper accordingly.
>
>
>
> > Training time, Loss curves
>
> **Ans**: Thank you for your suggestion. In response, we have included the training time (Tables 20–22) and loss curves (Figure 11) in Appendix E.7.
>
> The loss curves clearly demonstrate the model’s convergence.
>
> Regarding training time, the model requires the same number of steps as other GNN models. However, each step takes longer compared to classical GCN. For your convenience, we have also summarized the results here.
>
> We measured the average training time per step on the three largest datasets used in our experiments, varying |$\Omega$|. The training time ranges from 3x to 5x that of GCN, indicating reasonable time for convergence.
>
> |                | Penn94         | Cornell5       | Genius         |
> |----------------|----------------|----------------|----------------|
> | # Nodes        | 41,554         | 18,660         | 421,961        |
> | # Edges        | 1,362,229      | 790,777        | 984,979        |
>
> | **Penn94**         | **Training Time** | **Multiples of GCN** |
> |---------------------|--------------------|----------------------|
> | GCN                 | 0.0637     | -                    |
> | $\|\Omega\|$=2  | 0.2880       | 4.51x                |
> | $\|\Omega\|$=4   | 0.2888      | 4.53x                |
> | $\|\Omega\|$=8   | 0.2933      | 4.60x                |
> | $\|\Omega\|$=16   | 0.3007      | 4.71x                |
>
> | **Cornell5**       | **Training Time** | **Multiples of GCN** |
> |---------------------|--------------------|----------------------|
> | GCN                 | 0.0602     | -                    |
> | $\|\Omega\|$=2    | 0.2201      | 3.65x                |
> | $\|\Omega\|$=4    | 0.2056      | 3.41x                |
> | $\|\Omega\|$=8    | 0.2161      | 3.58x                |
> | $\|\Omega\|$=16    | 0.2332      | 3.87x                |
>
> | **Genius**         | **Training Time** | **Multiples of GCN** |
> |---------------------|--------------------|----------------------|
> | GCN                 | 0.1892      | -                    |
> | $\|\Omega\|$=2    | 0.8467     | 4.47x                |
> | $\|\Omega\|$=4    | 0.8911      | 4.70x                |
> | $\|\Omega\|$=8    | 0.9211      | 4.86x                |
> | $\|\Omega\|$=16    | 1.0756       | 5.68x                |

---

> > ### Comment · Reviewer_6te7 · 2024-11-26
> >
> > Thank you for your response.

---

### Comment · Area_Chair_dAhB · 2024-11-23

Dear Reviewers,

The authors have uploaded their rebuttal. Please take this opportunity to discuss any concerns you may have with the authors.

Best regards,
AC

---

### Meta-Review · Area_Chair_dAhB · 2024-12-21

**Metareview:**

This paper proposes a differentiable cluster-based GNN (DC-GNN) to address over-squashing and heterophily issues in graph neural networks. It integrates clustering into the message-passing mechanism, optimizing both node and cluster-node embeddings iteratively, and provides theoretical guarantees on convergence. The authors present experimental results demonstrating the model’s effectiveness across multiple datasets.

While the proposed DC-GNN model shows promising results, the reviewers have identified several weaknesses that need to be addressed:

1. The model's requirement for additional nodes and updates regarding clustering and assignment results in inefficiencies in both time and memory usage. This significantly limits its ability to handle large-scale datasets, making the trade-off between efficiency and performance unfavorable.
2. The paper lacks a comprehensive analysis of the convergence speed. The model may require more training rounds than initially expected. Additionally, no comparison of convergence behavior has been made between the proposed model and the baseline GNNs.
3. The experimental results are not convincing, as some baseline results are lower than commonly expected benchmarks. The model's improvement, particularly on heterophilic graphs, appears marginal. Additionally, the necessary artifact of the work is not submitted promptly for checking and reproduction.

Based on these weaknesses, we recommend rejecting this paper. We hope this feedback helps the authors improve their paper.

**Additional Comments On Reviewer Discussion:**

The authors' rebuttal efforts include addressing the reviewer's concerns about the model's hyperparameters, training time, and convergence. They also provide additional information on the baseline results and the model's performance on long-range graph datasets. However, during the rebuttal and discussion phase, the reviewers’ concerns regarding the scalability of the model and the experimental results remain unresolved. As a result, I recommend rejection based on the reviewers’ feedback.

---

### Decision · Program_Chairs · 2025-01-22

Reject